# Precision size and refractive index analysis of weakly scattering nanoparticles in polydispersions

Anna D. Kashkanova[1,2,5], Martin Blessing[1,2,3,5], André Gemeinhardt[1,2,3], Didier Soulat[4] and Vahid Sandoghdar[1,2,3 ✉]

Characterization of the size and material properties of particles in liquid suspensions is in very high demand, for example, in the analysis of colloidal samples or of bodily fluids such as urine or blood plasma. However, existing methods are limited in their ability to decipher the constituents of realistic samples. Here we introduce iNTA as a new method that combines interferometric detection of scattering with nanoparticle tracking analysis to reach unprecedented sensitivity and precision in determining the size and refractive index distributions of nanoparticles in suspensions. After benchmarking iNTA with samples of colloidal gold, we present its remarkable ability to resolve the constituents of various multicomponent and polydisperse samples of known origin. Furthermore, we showcase the method by elucidating the refractive index and size distributions of extracellular vesicles from *Leishmania* parasites and human urine. The current performance of iNTA already enables advances in several important applications, but we also discuss possible improvements.

Size and refractive index represent two key attributes of nanoparticles found in a wide range of disciplines such as medicine[1], pharmaceuticals[2], the food industry[3] and agriculture[4]. The samples of interest often consist of natural or synthetic suspensions of different origin and composition. For example, bodily fluids such as blood plasma, cerebro-spinal fluids or urine contain bioparticles such as extracellular vesicles (EVs), covering a large spectrum of size and protein or RNA content, which serve as disease markers[5,6]. Information about the size, material and abundance of particles in such heterogeneous mixtures is highly desirable in fundamental research as well as for clinical and industrial applications.

Various techniques can be employed to determine particle size distribution[7]. Electron microscopy (EM) provides exquisite resolution in direct imaging, but its sample preparation procedure, low speed and ex situ operation strongly hamper its appeal. Indeed, despite their intrinsic diffraction-limited resolution, optical methods dominate the diagnostic and analytical arena because they are fast and can be applied to a broad set of samples in the liquid phase. One of the workhorses in optical analysis is dynamic light scattering (DLS), which makes use of temporal correlations in the light scattered by an ensemble of diffusing particles[8]. In this method, the particle size is extracted from statistical analysis of the autocorrelation function of the light intensity. Today, DLS is the technique used most commonly in particle sizing, as it is easy to use and offers high accuracy. However, this method has a low size resolution, thus confronting limits in the analysis of polydisperse solutions[9]. A more recent approach, referred to as nanoparticle tracking analysis (NTA), analyzes the trajectories of individual particles to quantify their diffusion constants ($D$) and, thus, diameter $d$ (particles are considered spherical)[10,11]. Conventional NTA employs dark-field microscopy in which the signal is proportional to the scattering cross-section ($\sigma_{sca}$) of a particle and, thus, scales as $d^6$. This rapidly lowers the sensitivity of NTA for smaller particles. Currently, leading NTA instruments are validated for gold nanoparticles (GNP)

as small as 30 nm and polystyrene (PS) particles larger than 60 nm (refs. [11,12]). Furthermore, holography has been used for imaging and tracking particles, but the reported sensitivity corresponds to the scattering strength of PS particles with $d \approx 300$ nm (refs. [13–16]). Thus, there is a need for methods with higher sensitivity to access more weakly scattering particles and for better resolution in deciphering the constituents of heterogeneous mixtures. In accomplishing the latter goal, it is also very helpful to obtain valuable insight about the material composition of the particles under study. Indeed, the scattering signal in NTA has been used to assess the refractive index, but the precision in these studies has also been affected strongly by the limited signal-to-noise ratio (SNR) in dark-field microscopy.

In this work, we introduce iNTA as a new method that employs interferometric detection of scattering to analyze the trajectories and scattering cross-sections of diffusing single nanoparticles. Interferometric detection of scattering (iSCAT)[17,18], offers a highly efficient optical contrast, which has been used by a range of methods for label-free sensing of single proteins[19], mass photometry[20] and high-speed tracking of transmembrane proteins[21]. Here, we exploit the high SNR of iSCAT to achieve high precision in determining $D$, and thus the particle size $d$. In addition, we perform quantitative measurements of the iSCAT contrast to assess the scattering cross-section of each nanoparticle, providing direct information about its refractive index. We present unprecedented precision and resolution in measuring the size and the refractive index of nanoparticles, not only in monodisperse, but also in polydisperse mixtures of particles with diameter down to ~10 nm and of complex entities such as layered particles. Furthermore, we exhibit the advantages of iNTA in exemplary field applications such as the characterization of EVs from parasites and human urine.

## Results

**Measurement principle.** The diffusion of a particle in a fluid is described by the Stokes–Einstein (SE) equation

[1]Max Planck Institute for the Science of Light, Erlangen, Germany. [2]Max-Planck-Zentrum für Physik und Medizin, Erlangen, Germany. [3]Department of Physics, Friedrich-Alexander-Universität Erlangen-Nürnberg, Erlangen, Germany. [4]Institute of Clinical Microbiology, Immunology and Hygiene, Universitätsklinikum Erlangen and Friedrich-Alexander-Universität Erlangen-Nürnberg (FAU), Erlangen, Germany. [5]These authors contributed equally: Anna D. Kashkanova, Martin Blessing. ✉e-mail: vahid.sandoghdar@mpl.mpg.de

$$D = \frac{k_B T}{3\pi\eta d}, \qquad (1)$$

where $k_B$ is the Boltzmann constant, $T$ and $\eta$ are the temperature and viscosity of the fluid, respectively, and $d$ signifies the (apparent) diameter of the particle[22]. Thus, one can arrive at $d$ by evaluating $D$ from the mean squared displacement (MSD) of a particle trajectory. Because the number of trajectory points affects the measurement precision, fast recordings are highly desirable. However, high-speed imaging can only help if a large SNR is maintained to ensure low localization error[23]. This is where iSCAT microscopy provides a decisive advantage due to its ability to track nanoparticles with a high spatial precision and temporal resolution[18] (Methods).

Another quantity of importance in our work is the scattering cross-section $\sigma_{sca}$ of a nanoparticle. For uniform Rayleigh particles with $kd \ll 1$, where $k$ is the wavenumber, $\sigma_{sca} \propto |\alpha|^2$, where

$$\alpha = 3V \left( \frac{n_p^2 - n_m^2}{n_p^2 + 2n_m^2} \right) \qquad (2)$$

represents the polarizability[24]. Here, $V \propto d^3$ denotes the particle volume, and $n_p$ and $n_m$ are the refractive indices of the particle and its surrounding medium, respectively. The recorded iSCAT signal is proportional to the electric field of the scattered light and, thus, to $\alpha$[17]. It is expressed as a contrast ($C$) and can be read from the central interferometric point-spread function (iPSF) lobe (Methods). For particles with $kd \gtrsim 1$ and for multilayered particles, a generalized Mie theory describes the scattering strength (Supplementary Section 2.1). As we shall see, information on $C$ plays a decisive role in deciphering various species and determining their refractive indices in a polydispersion.

Figure 1a sketches a common wide-field setup for performing iSCAT measurements[18]. In the left column of Fig. 1b, we show three examples of the iPSF that result from the interference of planar (reflected from the sample interface) and spherical (scattered by the particle) waves[18,25]. The iPSFs vary qualitatively depending on the particle position relative to the coverslip and the focal plane[25]. To localize an iPSF in a given image, we apply radial variance transform (RVT), which converts the iPSF into a bright spot[26], as shown in the right column of Fig. 1b. An example of a trajectory is overlaid in Fig. 1c.

The details of sample preparation and the setup are described in the Methods and in Supplementary Section 1. Here, it suffices to note that we introduce a dilute suspension of nanoparticles in a closed chamber on a microscope coverslip (Fig. 1a) and image the diffusing entities using a fast camera. The focal plane of the microscope objective is placed approximately 1 μm above the coverslip and is stabilized with an active focus lock. The trajectory lengths are limited predominantly by the axial diffusion of particles.

We extract $D$ and thereby $d$ by fitting the MSD plot for individual trajectories. For monodisperse samples, we can also evaluate the mean diffusion constant $\bar{D}$ as well as a localization error by fitting averaged MSD plots, weighted by the trajectory length (see Supplementary Section 3.2 for details). For polydisperse samples, we additionally exploit the knowledge of $C$. Here, we report the maximum positive contrast from each trajectory because the interferometric contrast modulates in the axial direction as the particle traverses the illumination area[18] (Supplementary Sections 2.1, 2.2 and 3.1). We note that the common-path nature of our interferometric measurements makes them very robust against spurious phenomena that might affect the optical path.

**Monodisperse particle samples.** We start by applying iNTA to commercially available monodisperse samples to benchmark its performance. Figure 2 summarizes the outcome of our measurements on GNPs from two different manufacturers. The thin lines

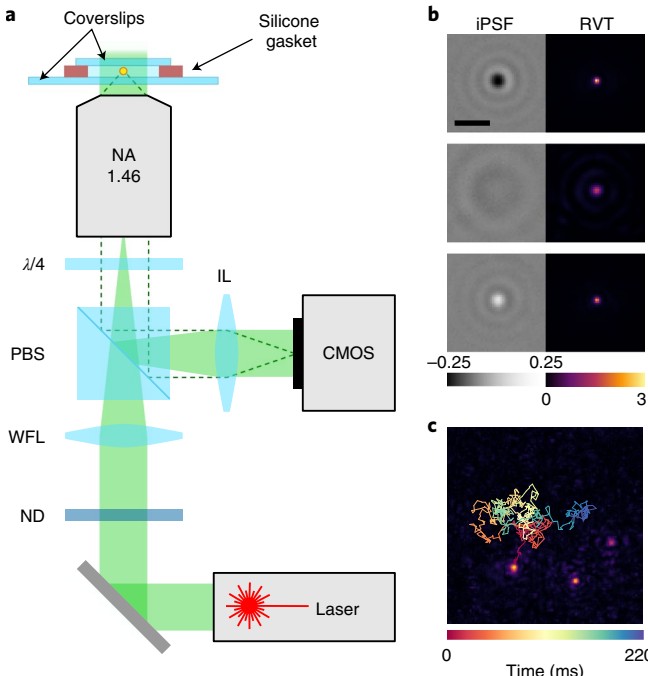

**Fig. 1 | iSCAT setup and trajectory extraction. a**, Wide-field iSCAT setup for tracking freely diffusing particles. Linearly polarized light from a laser traverses a neutral density filter (ND) used to adjust the incident power, passes a PBS) followed by a $\lambda/4$ plate that renders its polarization circular. A wide-field lens (WFL, $f = 400$ mm) focuses the light at the back focal plane of the objective. An imaging lens (IL, $f = 500$ mm) projects the reflected (solid green area) and scattered (dashed line) light on a CMOS camera chip. **b**, Three examples of iPSF (left) and the corresponding RVT images (right). Scale bar, 1 μm. **c**, A $7 \times 7$ μm² frame showing an RVT image of three 15 nm GNPs with an overlaid trajectory of one GNP recorded over 220 ms.

in Fig. 2a show MSD curves from individual trajectories that contained at least 25 localization events. The thick curves show the resulting linear averaged MSD plots, which confirm free diffusion.

Figure 2b shows the measured mean values $\bar{D}$ for GNPs of various diameters. The dashed line presents the behavior of the SE relation expected for the nominal average particle diameters ($\bar{d}_{nom}$) provided by the manufacturer. Although the agreement with the data is satisfactory (note the logarithmic scales), the high precision of our measurements reveals small deviations, which suggest a systematic correction to the particle size. Indeed, the solid curve in Fig. 2b reports an excellent agreement between theory and experiment if we consider an increase of the radius by $l_H = 1.8 \pm 0.3$ nm for all particles. We attribute the main effect to a hydration (stagnant) layer[27], but additional contributions might also come from surfactant molecules. To examine the SE relationship further, we also performed measurements at different temperatures using a micro heating stage (VAHEAT, Interherence). As exemplified for the case of 30 nm GNPs in the inset of Fig. 2b, we find agreement between the experimental data (symbols) and the prediction of equation (1) when replacing $\bar{d}_{nom}$ (dashed line) by a hydrodynamic diameter $d_H = \bar{d}_{nom} + 2l_H$ (solid line).

Figure 2c shows histograms of the measured diameters ($d_{mes}$) extracted from individual GNP trajectories. Gaussian fits to the data establish normal distributions, allowing us to determine a mean value $\bar{d}_{mes}$ and s.d. $\sigma_d^{(mes)}$ (ref. [28]). Table 1 presents these data as well as the uncertainty ($\Delta \bar{d}_{mes}$) in determining $\bar{d}_{mes}$. In addition, we list the extracted values of $l_H = (\bar{d}_{mes} - \bar{d}_{nom})/2$ and its error bars ($\Delta l_H$) for each measurement series (inset, Fig. 2c). We verified that the

measured quantities do not depend on the incident laser power, camera chip illumination or the focal plane position.

To compare iNTA with the existing state-of-the-art methods, we used DLS (ZetaSizer Nano ZS), NTA (Nanosight NS500), scanning electron microscopy (SEM, Hitachi S-4800) and transmission electron microscopy (TEM, Zeiss EM10) instruments to characterize GNPs with a nominal diameter of 30 nm, as an exemplary sample at the lower limit of the commercial NTA. The results of these measurements are summarized in Fig. 2d. It is evident that the DLS and NTA size distributions have larger spreads than those of SEM and TEM measurements. The width of the iNTA distribution, however, rivals that of TEM. This enables the measurement of highly resolved particle size distributions using an all-optical technique. We note that the systematic and technical errors encountered in iNTA are common to all NTA experiments and stem from uncertainties in temperature, drift and vibrations[29] (Supplementary Section 2.3). Furthermore, the accuracy for each method (that is, the mean value of the measured distribution) depends strongly on careful calibrations and consideration of various systematic effects.

**Polydisperse particle samples.** One of the central demands on sensing and sizing technologies is the identification of different species in a mixture. To investigate the performance of iNTA in such applications, we prepared various mixtures. First, we considered a suspension containing 15 nm, 20 nm, and 30 nm GNPs. To set the ground, in Fig. 3a, we show the number-weighted distribution of a DLS measurement (ZetaSizer Nano ZS), yielding a continuous featureless distribution. As shown in Fig. 3b, conventional NTA (Nanosight NS500) does not resolve the different populations either. In this case, we have also plotted a histogram of the scattering intensities along the right-hand vertical axis.

The scatter plot in Fig. 3c shows $\sqrt[3]{C}$ and $d_{mes}$ for individual trajectories extracted with iNTA. Motivated by the fact that $C \propto d^3$, the choice of $\sqrt[3]{C}$ conveniently provides a dimension on a par with $d$, and its value is in representing the scattering strength which, in turn, is related to the refractive index (equation (2) and Methods). A visual inspection of the data clearly reveals three clusters. In fact, the histogram of $\sqrt[3]{C}$ values plotted on the right-hand vertical axis also resolves the three populations on its own. Application of a two-dimensional (2D) Gaussian mixture model (GMM) with full covariance[30] lets us decipher the three populations in a quantitative manner and identify the populations in the $d_{mes}$ histogram. In Fig. 3d,e, we show other examples, where iNTA fully resolves

mixtures of 10 nm and 15 nm GNPs and of 40 nm and 60 nm PS spheres (PSS), respectively even though particles in this size range are usually not accessible in methods based on dark-field microscopy[31].

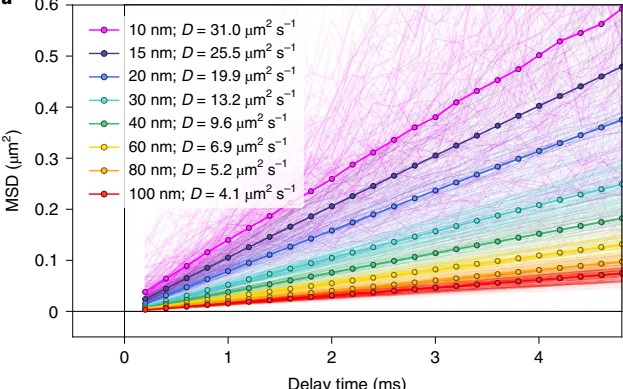

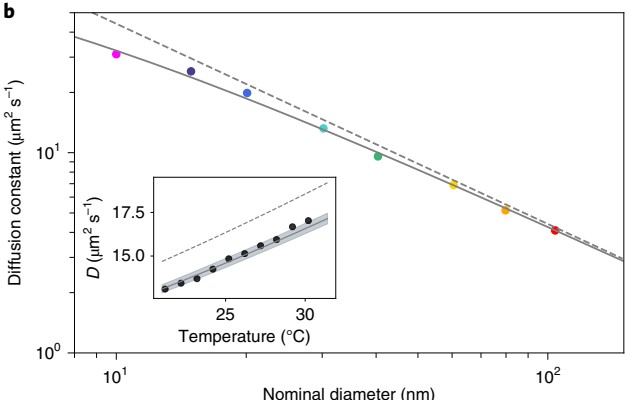

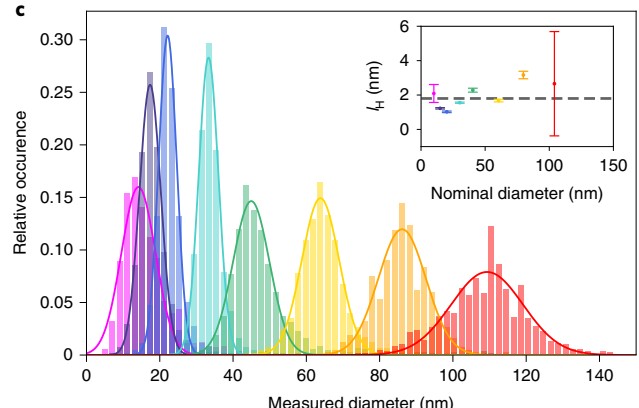

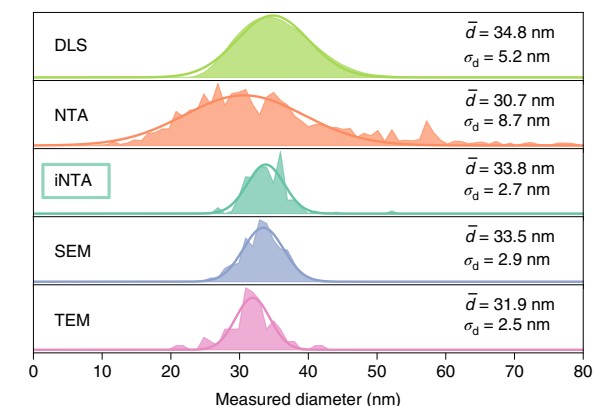

**Fig. 2 | Monodisperse particle samples. a**, MSD versus delay time for GNP samples of different sizes. Thin lines show the MSD extracted from each individual trajectory. Thick lines show the weighted average (by trajectory length). Diffusion constants extracted from the fits are listed in the legend. **b**, Diffusion constants extracted from the data in **a** versus the nominal GNP diameter provided by the manufacturer. Dashed gray line indicates the SE relation for $T = 21\,°C$. Solid line is a fit to the SE relation yielding an offset in particle radius by the hydration layer thickness of $l_H = 1.8 \pm 0.3$ nm. Inset: diffusion constants for 30 nm GNPs at different temperatures. The solid line shows the prediction of equation (1) for $l_H = 1.8$ nm. The shaded area indicates the 95% confidence interval for $l_H$ of $\pm 0.3$ nm. Dashed line shows the outcome of SE for $\bar{d}_{nom}$. **c**, Histograms of particle diameters extracted from the SE relation. Individual measurements were weighed by their trajectory lengths (Supplementary Section 3.2). The data for 10 nm, 15 nm, 20 nm and 30 nm GNPs were recorded at 40 mW illumination power; the rest were recorded at 2 mW. Inset: symbols show $l_H$ and its error bars as defined in Table 1. Dashed line indicates the value of $l_H$ obtained from the global fit in **b**. **d**, Comparison between different measurement techniques. The output of DLS measurements represents the number-weighted distribution.

**Table 1 | Nominal (nom) and measured (nes) properties of various GNP samples**

| Manufacturer | $\bar{d}_{nom}$ | $\Delta d_{nom}$ | $\sigma_d^{(nom)}$ | $\bar{d}_{mes}$ | $\Delta d_{mes}$ | $\sigma_d^{(mes)}$ | $l_H$ | $\Delta l_H$ | Number of trajectories | Number of localizations (×10⁶) |
|---|---|---|---|---|---|---|---|---|---|---|
| BBI | 10 | 1 | ≤1 | 14.2 | 0.3 | 4.7 | 2.1 | 0.6 | 819 | 0.04 |
| BBI | 14.9 | – | ≤1.5 | 17.4 | 0.1 | 2.9 | 1.3 | 0.0 | 11,341 | 1.28 |
| BBI | 20.1 | – | ≤1.6 | 22.1 | 0.1 | 2.5 | 1.0 | 0.0 | 6,635 | 1.35 |
| BBI | 30.2 | – | ≤2.4 | 33.2 | 0.1 | 2.6 | 1.6 | 0.1 | 1,535 | 0.83 |
| BBI | 40.4 | – | ≤3.2 | 45.0 | 0.2 | 5.0 | 2.3 | 0.1 | 2,943 | 1.24 |
| BBI | 60.5 | – | ≤4.8 | 63.9 | 0.2 | 5.1 | 1.7 | 0.1 | 2,401 | 1.48 |
| BBI | 79.8 | – | ≤6.4 | 85.9 | 0.4 | 6.6 | 3.2 | 0.2 | 1,234 | 1.15 |
| SA | 104 | 6 | ≤8 | 109.3 | 0.9 | 9.8 | 2.7 | 3.0 | 712 | 0.97 |

All quantities are in units of nanometers. See text for the definitions of $\bar{d}$ and $\sigma_d$. $\Delta d$ is computed as the 95% confidence interval of the mean by multiplying the standard error of the fit by 1.96 (ref. [28]). $\sigma_d^{(nom)}$ is calculated from the manufacturer (BBI Solutions, Sigma-Aldrich) specified coefficient of variation. $l_H$ represents the thickness of the hydration layer determined as $(\bar{d}_{mes} - \bar{d}_{nom})/2$. $\Delta l_H$ is the error on the hydration layer thickness calculated as $\sqrt{\Delta d_{nom}^2 + \Delta d_{mes}^2}/2$. The number of extracted trajectories as well as the total number of localizations (in millions) are indicated.

The advantage of combining the knowledges of $C$ and $d_{mes}$ becomes even more apparent when particles of similar size or $\sigma_{sca}$ are analyzed, as shown in Fig. 3f for a mixture of 40 nm GNPs, 100 nm PSS and 100 nm silica beads (SB). Neither $C$ nor $d_{mes}$ alone can provide clear information about the composition of the sample, but their combination in a 2D scatter plot provides very robust evidence for the existence of three different species. Again, as shown by the color-coded overlays, application of a GMM analysis allows us to decompose the $C$ and $d_{mes}$ histograms. We note in passing that the horizontal stretch of the data clouds in Fig. 3c,d is due to the uncertainty resulting from short trajectories or localization errors. The diagonal extension of the data in Fig. 3e,f, however, reveals the true size distribution in the sample.

Measurements of $C$ and $d_{mes}$ also provide direct access to the refractive index (RI) of the particles. Here, we considered RI for gold ($n_{Au} = 0.63 + 2.07i$)[32] and fitted the data in Fig. 3c,d,f, resulting in the solid lines. The intercept of the horizontal axis yields another independent measure for the hydration shell $2l_H$, which amounts to 1.6 nm, 1.8 nm and 1.5 nm for the three cases, respectively. We used this information to relate the experimentally measured $C$ and the expected value of $\sigma_{sca}$ with one single calibration parameter for our setup (Supplementary Section 2.1). Next, we use this calibration and fit the data for PSS and SB in Fig. 3e,f to arrive at $n_{PS} = 1.62$ and $n_{Si} = 1.45$, which are in good agreement with the literature values signified by the gray bands[33–35]. We remark that for PSS and SB, the RI curves calculated from the full Mie theory[24] deviate from a straight line because $\sigma_{sca}$ for larger particles begins to contain contributions from higher order multipoles.

We also investigated a more complex mixture of 10 nm and 20 nm GNPs with and without polyethylene glycol coatings (Creative Diagnostics; molecular weight of PEG 3,000). Figure 3g shows the high performance of iNTA by clearly distinguishing four populations. Moreover, the measurements provide us with the direct assessment of the thickness of the PEG layer, which in this case corresponds to about 12 nm. These results pave the way for future sensitive and quantitative investigation of composite nanoparticles and their interaction with the surrounding liquid phase[36].

The superior sensitivity and resolution of iNTA measurements on monodisperse and known polydisperse nanoparticles prompted us to employ it in realistic field problems. Indeed, there is a substantial number of applications in which nanoparticles of various substances and sizes need to be characterized in a fast, accurate and noninvasive manner. Here, we discuss the analysis of synthetically produced lipid vesicles as well as EVs, which contain various proteins, nucleic acids or other biochemical entities either in their interior or attached to them. EVs have been identified as conveyers for cell–cell communication and as disease markers[5,6], but studies are

partly hampered by the throughput and resolution in their quantitative assessment[37,38]. EVs are often grouped as exosomes (diameter 30–150 nm, originating from inside a cell) and microvesicles (diameter 100–1,000 nm, stemming from the cell membrane), whereas particles smaller than 150 nm might also be referred to as small EVs (sEVs)[39,40]. We now discuss three case studies. Information about sample preparation and measurement conditions is found in the Supplementary Information.

**Synthetically produced liposomes.** Figure 4a shows the outcome of iNTA measurements on a sample of synthetically produced liposomes. To emphasize the small size regime that is not available to other methods such as holographic microscopy[13,15,16] and to ensure unilamellarity of the liposomes, we prepared them with a fraction of charged lipids (Supplementary Section 1.2) and extrusion through a 200 nm membrane[41]. Liposomes consist of a lipid bilayer shell surrounding an aqueous interior (inset, Fig. 4a) and can, therefore, be modeled by a generalized Mie theory[24] that takes into account the thickness ($t_{sh}$) and RI ($n_{sh}$) of the shell as well as the RI of the interior ($n_{in}$). The orange curve in Fig. 4a shows the result of fitting the data using Mie theory. If we assume the published value of $n_{sh} = 1.48$ for the lipids used in fabricating our liposomes[42], we deduce $t_{sh} = 5.7$ nm and $n_{in} = 1.334$ consistent with the RI of water.

We now examine the tolerance with which one could extract quantitative information about $t_{sh}$, $n_{sh}$ and $n_{in}$ from data such as in Fig. 4a with no previous knowledge of the lipids. In Fig. 4b, we plot the calculated isosurfaces of constant $C$ and $d_{mes}$ as a function of these parameters for three points marked in Fig. 4a. We find that all three surfaces meet when $n_{in} \simeq 1.334$ for water. In other words, choosing a different $n_{sh}$ would yield another $t_{sh}$ value but the same $n_{in}$. The range of possible $t_{sh}$ and $n_{sh}$ values, which is consistent with literature values for lipids, highlights the difficulty in their full characterization[43].

To illustrate the sensitivity of the results to variations of the liposome inner part, we plot the dashed curves in Fig. 4a for larger $n_{in}$ but the same shell parameters ($t_{sh} = 5.7$ nm and $n_{sh} = 1.48$). The graph clearly shows that even a slight change in $n_{in}$ can be detected reliably. We also note that the high SNR of iNTA lets us clearly discriminate against a simple model for a uniform nanosphere (Supplementary Section 4.1). We note that DLS and NTA measurements on this sample result in similar size distribution, but they do not yield any detailed information about the refractive index (Supplementary Section 4.2). Having established the ability of iNTA to provide insight into the composition of synthetic liposomes, we now consider two unknown biological samples.

**Parasite EVs.** Leishmaniasis is a potentially lethal disease, classified by the World Health Organization as 1 of the 20 neglected

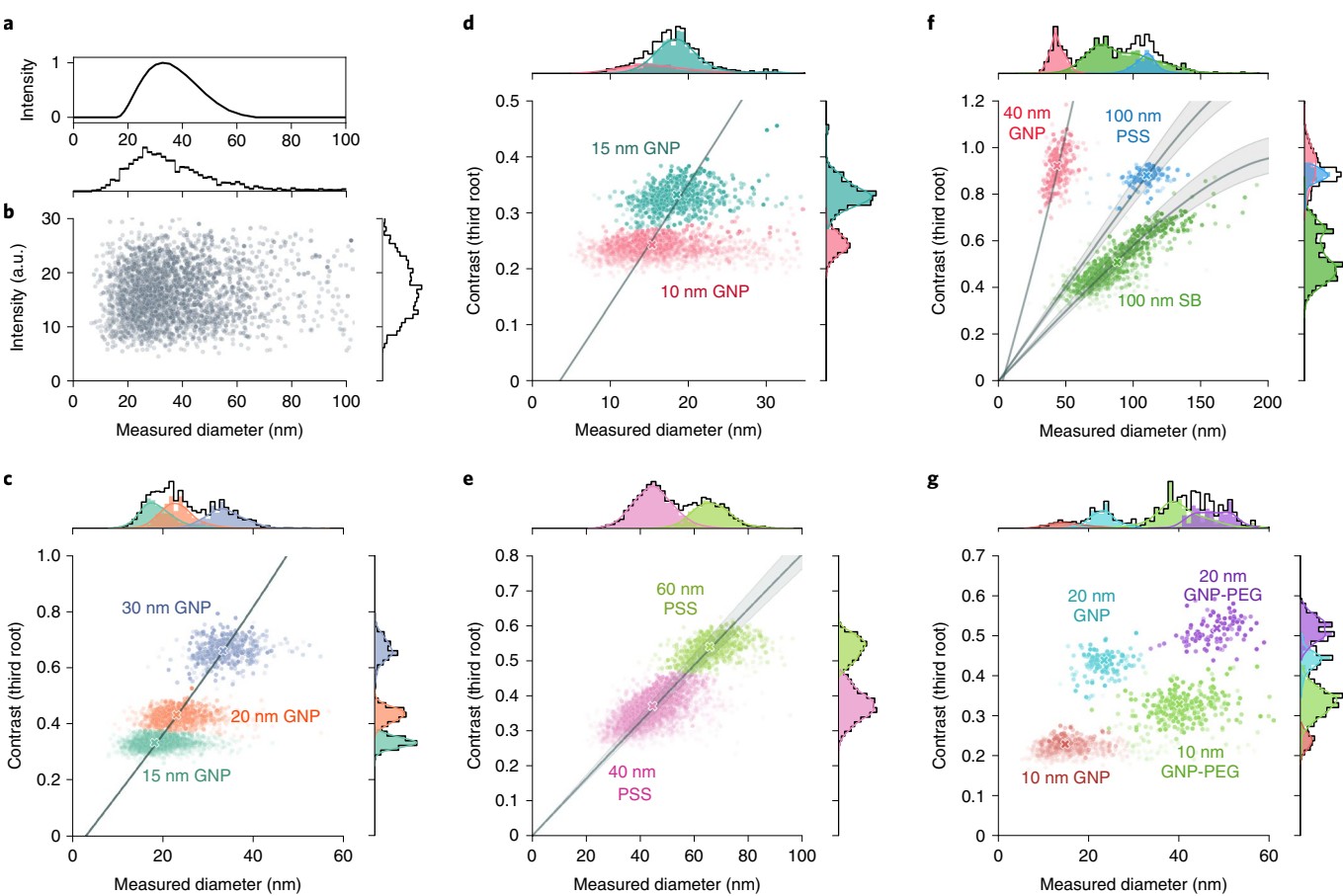

**Fig. 3 | Polydisperse particle samples. a–c**, DLS (**a**), NTA (**b**) and iNTA (**c**) measurements of a mixture of 15 nm, 20 nm and 30 nm GNPs. **d–g**, iNTA measurements of various mixtures as labeled in each graph. The horizontal and vertical axes denote the measured diameter and the third root of the iSCAT contrast, respectively. The transparency of each datapoint indicates the length of its trajectory. In each panel, a 2D GMM is used to identify different populations highlighted in color. The gray curves establish the relationship between $\sqrt[3]{C}$ and $d_{\text{mes}}$ according to the respective refractive indices and the shaded regions indicate the uncertainties in the refractive index data found in the literature: silica refractive index between 1.43 and 1.48, polystyrene refractive index between 1.58 and 1.68. Crosses in **c–g** signify the medians of each data cloud.

tropical diseases worldwide. *Leishmania* parasites secrete numerous virulence factors, most of which are carried together with small RNA and proteins inside EVs[44,45]. Quantitative characterization of the vesicles emitted by *Leishmania* would be of great value for understanding their role in the infection process, but reliable data are missing.

To investigate this system with our method, we enriched EVs that were secreted by Leishmania parasites in a culture medium and prepared them following a sequential centrifugation protocol (Supplementary Section 1.3). The quality of the EVs obtained in this fashion has been validated previously by electron microscopy[44]. In Fig. 4c, we present an iNTA scatter plot of the *Leishmania* EVs. The resulting size histogram is consistent with recently published DLS and NTA signals obtained with similar EV samples[46]. However, the iNTA data gives access to a more quantitative analysis of the EV heterogeneity.

Figure 4d shows the isosurfaces of the iSCAT contrast in the space spanned by $t_{\text{sh}}$, $n_{\text{sh}}$ and $n_{\text{in}}$ for three points marked in Fig. 4c. The ranges of $n_{\text{in}}$ and $n_{\text{sh}}$ are influenced by the amount of protein that is contained in the aqueous inner volume and in the lipid shell, respectively. Moreover, the effective shell thickness $t_{\text{sh}}$ can be affected by the protein content. Interestingly, the three planes in Fig. 4d again cross along a single curve at $n_{\text{in}} = 1.363$, hinting to the likelihood that this is a robust common value. Indeed, if we assume $t_{\text{sh}} = 5$ nm as a reasonable thickness for a single lipid

bilayer, a fit given by the central green curve in Fig. 4c traversing through the three chosen points reproduces the data trend quite well, yielding $n_{\text{sh}} = 1.44$.

The larger spread in $C$ as compared with the data for synthetically produced liposomes (Fig. 4a) highlights the ability of iNTA to detect slight variations of $t_{\text{sh}}$, $n_{\text{sh}}$ or $n_{\text{in}}$. For instance, if we assume that the observed distribution of contrast results from variations in $n_{\text{in}}$, we estimate the protein content of the EV inner solution to be about 10% ± 3% as delineated by the solid olive curves, assuming an effective RI of 1.6 for protein matter[47].

**Human urine EVs.** Urine is also known to contain EVs, and scientists believe that these hold great promise to serve as disease markers[48,49]. Urine has been analyzed by NTA[35,50], yielding a broad unimodal vesicle size distribution in the range of 50–300 nm (Supplementary Fig. 17). However, Fig. 4e shows that, when following the same sample preparation procedure, the iNTA scatter plot clearly resolves various subpopulations. To examine the apparent data bifurcation at about $d = 100$ nm further, we applied proteinase K to the sample. This resulted in the elimination of the lower branch of the data cloud (teal), indicating the non-EV protein agglomerate nature of these nanoparticles (Supplementary Section 4.3). This finding highlights a clear advantage of the higher resolution of iNTA for identifying sample impurities—an issue that poses challenges in EV research[51]. We note that, as discussed in Supplementary

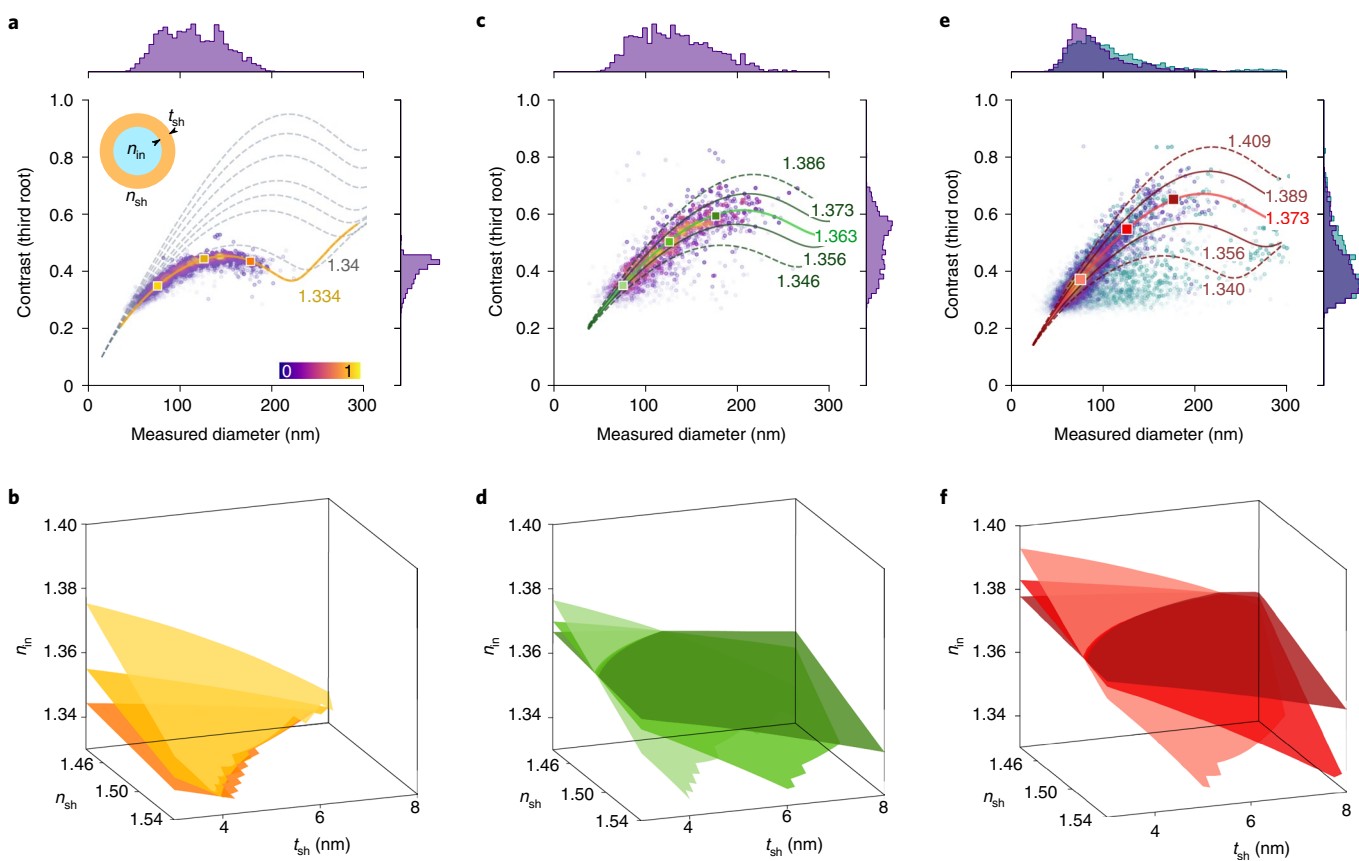

**Fig. 4 | Extracellular vesicles. a**, iNTA scatter plots of synthetically produced liposomes. Color bar denotes point density. The gray dashed contours correspond to different refractive indices of the inner part of the liposomes, starting at 1.334 (water) and increasing in steps of 0.02 from 1.34 for $n_{sh} = 1.48$ and $t_{sh} = 5.7$ nm. Inset: cartoon of a vesicle. **b**, Isosurfaces of constant particle size and iSCAT contrast for the points marked in **a** over a range of values for $n_{in}$, $n_{sh}$ and $t_{sh}$. **c**, iNTA scatter plots of EVs from *Leishmania* parasites. The green line indicates the best fit value for $n_{in}$, while the solid (dashed) olive lines indicate the 25th and 75th (10th and 90th) percentiles of the extracted $n_{in}$. **d**, Same as **b** but for the points marked in **c**. **e**, iNTA scatter plots of EVs from urine of a healthy human donor. The red line indicates the best fit value for $n_{in}$, and the solid (dashed) maroon lines indicate the 25th and 75th (10th and 90th) percentiles of the extracted $n_{in}$. **f**, Same as **b** but for the points marked in **e**. Each plot shows the outcome of one iNTA measurement performed on one sample. The results from more samples are shown in Supplementary Fig. 18.

Section 4.3, we performed several control experiments to verify that our urine samples indeed contained EVs according to published guidelines[51].

As presented in Fig. 4f, the iNTA data from urine EVs could also be analyzed in the same manner as those from *Leishmania* to provide a quantitative sense of the tolerances in the set of parameters $n_{sh}$, $n_{in}$ and $t_{sh}$. Interestingly, again we find that isosurfaces of contrast for the three marked points share a common section at $n_{in} = 1.373$. We thus estimate the protein content in the inner solution to be 13% ± 6% as shown by the solid maroon curves in Fig. 4e.

## Discussion and outlook

We have introduced iNTA as an all-optical method that pushes the limits of sensitivity, precision and resolution in determining the size and refractive index of nanoparticle mixtures. The advantage of iNTA over existing techniques was demonstrated in several examples, such as detection of weakly scattering nanoparticles, gaining insight into the hydration layer of colloidal GNP, deciphering complex nanoparticle species in various polydispersions and characterizing EV. While in this work we have focused on such challenging tasks to demonstrate the added value of iNTA, the technique can also detect particles with larger signals, as has been reported by earlier interferometric techniques[13] (for limitations, see Supplementary Section 2.4). The iNTA measurement times are generally comparable

with those of DLS and NTA, and they could vary from a few seconds to the order of 10 min, depending on the sample concentration, particle sizes and RIs in the sample as well as the desired precision (Supplementary Section 2.5). Being based on single-particle tracking, the technique works with dilute samples, making it suitable for the detection of rare events and species.

We believe that the combination of the exceptional sensitivity and resolution, noninvasive nature, ease of sample preparation, low sample volume requirements and short measurement times of iNTA makes it suitable for a large number of applications[1–4,38]. Furthermore, the method can be improved by several means, for example, the use of a shorter laser wavelength and higher laser powers to increase the SNR as well as the employment of particle confinement strategies[52] to extend the measurement time. Moreover, iNTA can be automated readily for high throughput inspections and be combined with sensitive fluorescence measurements to extract additional information about the particles under study (Supplementary Fig. 16).

The high-resolution 2D scatter plots presented in this work serve to detect slight variations and anomalies among different samples. To assess the potential of this capability for medical diagnosis, for example, in the analysis of bodily fluids such as blood and urine, extensive studies are necessary to first catalog the range of inhomogeneities in samples from healthy donors. In addition, we consider an exciting application of iNTA to be in fundamental research of

biological processes that involve cellular nanoentities such as protein condensates and lipid droplets[53] as well as products of cellular secretion such as EVs[38].

## Online content

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

## Methods

**iSCAT microscopy.** The iSCAT intensity recorded on the detector reads

$$I_{det} \propto |E_{ref}|^2 + |E_{sca}|^2 + 2|E_{ref}||E_{sca}| \cos\theta, \tag{3}$$

where $I_{ref} = |E_{ref}|^2$ and $I_{sca} = |E_{sca}|^2$ denote the intensities of the reference and the scattered light, respectively. The phase $\theta$ stands for the relative phase between the two fields, which can arise from a Gouy phase in the imaging system, material-dependent scattering phase and a traveling phase component stemming from the axial position of the particle. This expression is very similar to the signal in holography. However, contrary to the conventional use of holography in imaging, iSCAT works have employed interferometry to detect weak signals from nanoparticles[18]. An important feature that has made this possible is using a common-path arrangement, which is usually implemented in the reflection mode[18]. The iSCAT contrast is defined as

$$C = \frac{I_{det} - I_{bg}}{I_{bg}}, \tag{4}$$

where $I_{bg}$ is the intensity of the background in the vicinity of the particle image. Therefore, in the limit of $|E_{sca}| \ll |E_{ref}|$, $C$ is proportional to $|E_{sca}/E_{ref}|$. For a Rayleigh particle $E_{sca}$ is proportional to $\alpha$. Hence, we find that $C \propto \alpha \propto V \propto d^3$.

**Measurement setup.** Figure 1a shows a diagram of our iSCAT microscope. A low-coherence light source with a wavelength of 525 nm (Lasertack) is focused onto the back focal plane of a 63× oil immersion objective (numerical aperture 1.46, Zeiss). A neutral density (ND) filter is used to adjust the laser power. A $\lambda/2$ waveplate located directly after the ND filter (not shown in Fig. 1a) is used to match the polarization of the incident light to be transmitted through the polarization-dependent beam splitter (PBS). A $\lambda/4$ waveplate changes the polarization of the light from linear to circular. Circularly polarized light is partially reflected by the coverslip and partially scattered by the particles, reversing its handedness. On going through the $\lambda/4$ waveplate, the polarization changes back to linear, but now rotated by 90°. The light is reflected by the PBS towards the complementary metal oxide semiconductor (CMOS) camera chip (Photon Focus MV1-D1024E-160-CL-12). We typically use a field of view (FoV) of 128 pixels × 128 pixels, equivalent to 7 × 7 μm². The recording speed is 5,000 frames per second (fps), limited by the camera readout time. By recording 1-s-long videos, we capture trajectories with more than 100 localizations even for 10 nm particles and with more than 1,000 localizations for particles larger than 20 nm (Supplementary Fig. 8).

**Measurement procedure.** We set the exposure to the maximum possible value of $t_{exp} = 80\,\mu s$ for the frame rate of 5,000 fps. A total of 200 or 600 1-s-long videos were recorded for monodisperse and polydisperse samples, respectively. More videos (1200) were recorded for the liposome and urine samples as they were fairly dilute. For monodisperse sample measurements, we use an image trigger built into the video acquisition software (pyLabLib Cam-control) to start saving the frames 0.5 s before the particle crosses the center of the FoV. For polydisperse solution measurements, we do not use a particle trigger but rather record a video every few seconds or continuously. To adjust the focal plane position above the coverslip, we first focus on the coverslip and then use the piezo-electric stage to position the focal plane at a necessary position (at around 1 μm) above the coverslip. To lock the focus position, we used a position sensing detector (PDP90A) combined with a red laser operating in TIR mode (CPS670F) and a PSD auto aligner (TPA101). All components were purchased from Thorlabs Inc.

**Trajectory statistics.** The following number of trajectories/localizations were included in the figures presented in this manuscript.

In Fig. 2b (inset) each point is calculated from around 3,000 trajectories comprising $1 \times 10^6$ localizations. In Fig. 2d, NTA measurements include 1,156 trajectories comprising 12,932 localizations, while iNTA measurements include 50 trajectories comprising 27,767 localizations. The SEM histogram includes 324 particles, whereas the TEM histogram includes 87 particles. The minimum trajectory length for iNTA measurements was always set to 25 points. In Fig. 3b–f, the number of trajectories (localizations) plotted in each panel are, respectively, 2,516 (20,694); 3,140 ($1.30 \times 10^6$); 2,292 ($0.33 \times 10^6$); 3,805 ($2.2 \times 10^6$); 1,162 ($1.30 \times 10^6$) and 967 ($0.42 \times 10^6$). For iNTA measurements, the minimal trajectory length was set to 100 localizations for (Fig. 3c,e,f) and 25 localizations for (Fig. 3d,g). In Fig. 4a,c,e the number of trajectories (localizations) in each panel are 2,431 ($2.3 \times 10^6$), 2,110 ($2.25 \times 10^6$), 7,003 ($4.03 \times 10^6$) in the violet data and 6,277 ($4.49 \times 10^6$) in the teal data. The minimum trajectory length was set to 100 localizations.

**Ethics oversight.** The Ethics Committee of the Friedrich Alexander University Erlangen waived the need for ethics approval. Written informed consent was obtained from the urine donor.

**Reporting Summary.** Further information on research design is available in the Nature Research Reporting Summary linked to this article.

## Data availability

An example raw dataset is available in the iNTA repository (https://github.com/SandoghdarLab/iNTA). Considering the large size of the individual raw videos, more data are available from the corresponding author on request. We have also submitted all relevant data to the EV-TRACK knowledge base (EV-TRACK ID: EV220073)[54].

## Code availability

Video acquisition was performed using a custom pyLabLib Cam-control software developed in our laboratory and written in Python v.3.6 (https://github.com/SandoghdarLab/pyLabLib-cam-control). The software was not developed by the authors of this manuscript, and is not central to the research described here. In principle, any video acquisition software can be used, as long as acquisition at high frame rates is supported. The main data analysis pipeline was written in Python v.3.6 and uses standard Python packages as well as imgrvt v.1.0.0, pyLabLib v.1.2.1, trackpy v.0.5.0. It is available in the iNTA repository (https://github.com/SandoghdarLab/iNTA) as v.1.0. It will also be incorporated into the PiSCAT package (https://github.com/SandoghdarLab/PiSCAT). In principle, any particle tracking software can be used after application of Radial Variance Transform (https://github.com/SandoghdarLab/rvt) to the median background corrected videos. The Mie scattering calculations were performed in Mathematica v.10.0, Matlab v.2019a and Python v.3.6.

## References

54. Deun, J. V. et al. EV-TRACK: transparent reporting and centralizing knowledge in extracellular vesicle research. *Nat. Methods* **14**, 228–232 (2017).

## Acknowledgements

We are grateful to M. Reischke for assistance with iNTA measurements, A. Shkarin for the video acquisition software, A. Schambony and A. Giessl for TEM measurements, E. Butzen for SEM measurements, A. Eigen (Halik laboratory, FAU) for support with DLS measurements and A. Zika (Gröhn laboratory, FAU) for support with NTA measurements. We also thank F. Gröhn, K.-U. Eckardt, P. Enghard, R. Böckmann, M. Bonn, J. van Deun, A. Schambony, D. Albrecht, M. Mazaheri and K. Kasaian for helpful discussions. We are grateful to M. Küppers, A. Shkarin and J. Lühr for careful reading of the manuscript and insightful comments. We thank the Max Planck Society and Alexander von Humboldt Foundation (fellowship for A.D.K.) for financial support.

## Author contributions

A.D.K., M.B., A.G. and V.S. conceived the experiments, M.B. performed the experiments, A.D.K., M.B. and A.G. analyzed the data. D.S. contributed materials. All authors discussed the results and commented on the manuscript. A.D.K., M.B., A.G. and V.S. wrote the paper. V.S. supervised the project.

## Funding

## Competing interests

A.D.K., M.B., A.G. and V.S. have filed an International Patent Application (PCT) based on this work in the name of the Max Planck Gesellschaft zur Förderung der Wissenschaften e.V. D.S. declares no competing interests.

## Additional information

**Correspondence and requests for materials** should be addressed to Vahid Sandoghdar.

# Reporting Summary

## Statistics

For all statistical analyses, confirm that the following items are present in the figure legend, table legend, main text, or Methods section.

| n/a | Confirmed | |
|---|---|---|
| ☐ | ☒ | The exact sample size (*n*) for each experimental group/condition, given as a discrete number and unit of measurement |
| ☐ | ☒ | A statement on whether measurements were taken from distinct samples or whether the same sample was measured repeatedly |
| ☒ | ☐ | The statistical test(s) used AND whether they are one- or two-sided<br>*Only common tests should be described solely by name; describe more complex techniques in the Methods section.* |
| ☒ | ☐ | A description of all covariates tested |
| ☒ | ☐ | A description of any assumptions or corrections, such as tests of normality and adjustment for multiple comparisons |
| ☐ | ☒ | A full description of the statistical parameters including central tendency (e.g. means) or other basic estimates (e.g. regression coefficient) AND variation (e.g. standard deviation) or associated estimates of uncertainty (e.g. confidence intervals) |
| ☒ | ☐ | For null hypothesis testing, the test statistic (e.g. *F*, *t*, *r*) with confidence intervals, effect sizes, degrees of freedom and *P* value noted<br>*Give P values as exact values whenever suitable.* |
| ☒ | ☐ | For Bayesian analysis, information on the choice of priors and Markov chain Monte Carlo settings |
| ☒ | ☐ | For hierarchical and complex designs, identification of the appropriate level for tests and full reporting of outcomes |
| ☒ | ☐ | Estimates of effect sizes (e.g. Cohen's *d*, Pearson's *r*), indicating how they were calculated |

*Our web collection on statistics for biologists contains articles on many of the points above.*

## Software and code

Policy information about availability of computer code

| Data collection | Video acquisition was performed using a custom pyLabLib Cam-control software developed in our lab and written in Python 3.6 (https://github.com/SandoghdarLab/pyLabLib-cam-control). The software was not developed by the authors of this manuscript, and is not central to the research described here. In principle any video acquisition software can be used, as long as acquisition at high frame rates is supported. |
|---|---|
| Data analysis | The main data analysis pipeline was written in Python 3.6 and uses freely available standard Python packages as well as imgrvt 1.0.0, pyLabLib 1.2.1, trackpy 0.5.0. It is available in the iNTA repository (https://github.com/SandoghdarLab/iNTA) as version 1.0. It will be incorporated into the PiSCAT package (https://github.com/SandoghdarLab/PiSCAT). In principle any particle tracking software can be used after application of Radial Variance Transform (https://github.com/SandoghdarLab/rvt) to the median background corrected videos. The Mie scattering calculations were performed in Mathematica 10.0, Matlab 2019a and Python 3.6. |

For manuscripts utilizing custom algorithms or software that are central to the research but not yet described in published literature, software must be made available to editors and reviewers. We strongly encourage code deposition in a community repository (e.g. GitHub). See the Nature Portfolio guidelines for submitting code & software for further information.

## Data

Policy information about availability of data

All manuscripts must include a data availability statement. This statement should provide the following information, where applicable:
- Accession codes, unique identifiers, or web links for publicly available datasets
- A description of any restrictions on data availability
- For clinical datasets or third party data, please ensure that the statement adheres to our policy

An example raw dataset is available in the iNTA repository (https://github.com/SandoghdarLab/iNTA). Due to large size of the individual raw videos, more data are

# Field-specific reporting

Please select the one below that is the best fit for your research. If you are not sure, read the appropriate sections before making your selection.

☒ Life sciences ☐ Behavioural & social sciences ☐ Ecological, evolutionary & environmental sciences

For a reference copy of the document with all sections, see nature.com/documents/nr-reporting-summary-flat.pdf

# Life sciences study design

All studies must disclose on these points even when the disclosure is negative.

| | |
|---|---|
| Sample size | We measure nanoparticles diffusing in solutions and each presented histogram contains on the order of 1000 trajectories and 1e6 localization with details for each individual measurement given in the Methods section. This number of trajectories and localizations was a result of keeping measurement time reasonable (on the order of several minutes) for dilute samples. |
| Data exclusions | Only trajectories with more than 25 localizations were kept to decrease the error in diffusion constant determination. |
| Replication | The measurements are very reproducible. We verified that the measured hydrodynamic size does not depend on the incident laser power, camera chip illumination, and the focal plane position. However as indicated in the inset of figure 2b, temperature and viscosity have an effect on the extracted diffusion constant and therefore need to be controlled. Additionally the buffer properties (pH, salt concentration) could have an effect on the hydration layer and therefore measured particle size. As is now shown in the SI selected measurements were repeated 3-20 times to ensure repeatability. |
| Randomization | Not applicable. The measurement was by default performed on a random sample of particles, which diffuse into the illumination volume. The necessary amount of solution was also selected randomly with a pipette. |
| Blinding | Not required, since all the samples of the same kind (e.g. all biological samples) were analyzed with the same exact code without modifications. In that sense the analysis is "blind". |

# Reporting for specific materials, systems and methods

We require information from authors about some types of materials, experimental systems and methods used in many studies. Here, indicate whether each material, system or method listed is relevant to your study. If you are not sure if a list item applies to your research, read the appropriate section before selecting a response.

## Materials & experimental systems

| n/a | Involved in the study |
|---|---|
| ☐ | ☒ Antibodies |
| ☒ | ☐ Eukaryotic cell lines |
| ☒ | ☐ Palaeontology and archaeology |
| ☐ | ☒ Animals and other organisms |
| ☐ | ☒ Human research participants |
| ☒ | ☐ Clinical data |
| ☒ | ☐ Dual use research of concern |

## Methods

| n/a | Involved in the study |
|---|---|
| ☒ | ☐ ChIP-seq |
| ☒ | ☐ Flow cytometry |
| ☒ | ☐ MRI-based neuroimaging |

## Antibodies

| | |
|---|---|
| Antibodies used | Primary antibodies<br><br>anti-TSG101: Proteintech [Manchester, UK]; 28283-1-AP; NaN; 00080154<br>anti-Flotilin-1: Proteintech [Manchester, UK]; 15571-1-AP; NaN; 00053543<br>anti-Uromodulin: Proteintech [Manchester, UK]; 11911-1-AP; NaN; 00056125<br>anti-CD63: Proteintech [Manchester, UK]; 67605-1-Ig; 3D4D1; 10015952<br>anti-CD81: Proteintech [Manchester, UK]; 66866-1-Ig; 1G2C6; 10017531 & 10011817<br>anti-Alix: Proteintech [Manchester, UK]; 67715-1-Ig; 1H9D9; 10017670<br><br>Secondary antibodies<br><br>Goat Anti-Mouse IgG (H+L)-HRP Conjugate; Bio-Rad Laboratories [Feldkirchen, GER]; #170-6516; 64322731<br>Goat Rabbit-Mouse IgG (H+L)-HRP Conjugate; Bio-Rad Laboratories [Feldkirchen, GER]; #170-6515; 64332296 |

| Validation | See the attached Antibody_validation.pdf |
|---|---|

## Animals and other organisms

Policy information about <u>studies involving animals</u>; <u>ARRIVE guidelines</u> recommended for reporting animal research

| Laboratory animals | The study did not involve laboratory animals |
|---|---|

| Wild animals | The study did not involve wild animals |
|---|---|

| Field-collected samples | Leishmania exosomes were purified from axenic culture of Leishmania major promastigotes (strain MHOM/IL/81/ FEBNI) |
|---|---|

| Ethics oversight | No ethical approval or guidance was required, since Leishmania promastigotes are not live vertebrates. |
|---|---|

Note that full information on the approval of the study protocol must also be provided in the manuscript.

## Human research participants

Policy information about <u>studies involving human research participants</u>

| Population characteristics | One white male, 30 years old |
|---|---|

| Recruitment | Only one volunteer was recruited. This prevents us from making general statements about human urine, beyond this particular sample. However, a single sample was sufficient to demonstrate the advantages of iNTA. |
|---|---|

| Ethics oversight | The Ethics Committee of the Friedrich Alexander University Erlangen waived the need for ethics approval. The donor provided written informed consent prior to enrolment in the study. |
|---|---|

Note that full information on the approval of the study protocol must also be provided in the manuscript.

