## [Peer Review File · Nature Methods]

Peer Review Information

Manuscript Title: "Precision size and refractive index analysis of weakly scattering nanoparticles in polydispersions"

Corresponding author name(s): Vahid Sandoghar

Reviewer Comments & Decisions:

EA, duplicate for each version as needed, and then delete this instruction.

Decision Letter, initial version:

Dear Vahid,

Your Article entitled "Precision size and refractive index analysis of weakly scattering nanoparticles in polydispersions" has now been seen by two reviewers, whose comments are attached. While they find your work of potential interest, they have raised serious concerns which in our view are sufficiently important that they preclude publication of the work in Nature Methods, at least in its present form.

As you will see, the reviewers raise concerns about the level of benchmarking, the quality of the EV experiments, and whether the method really enables new and meaningful observations.

For Nature Methods to publish the piece, it needs to make a strong case that iNTA should replace standard NTA for exosome/EV research and become a gold standard for particle analysis (for biologists).

Should further experimental data allow you to fully address these criticisms we would be willing to look at a revised manuscript (unless, of course, something similar has by then been accepted at Nature Methods or appeared elsewhere). This includes submission or publication of a portion of this work somewhere else. We hope you understand that until we have read the revised paper in its entirety we cannot promise that it will be sent back for peer-review.

If you are interested in revising this manuscript for submission to Nature Methods in the future, please contact me to discuss your appeal before making any revisions. Otherwise, we hope that you find the reviewers' comments helpful when preparing your paper for submission elsewhere.

Sincerely,

Rita

Rita Strack, Ph.D.
Senior Editor
Nature Methods

Reviewers' Comments:

Reviewer #1:
None

Reviewer #2:

Remarks to the Author:

Kashkanova et al. combine two existing methods that are routinely used in the nanoparticle field, namely interferometric detection of scattering and nanoparticle tracking analysis, to identify both size and refractive index with a higher sensitivity and precision in a single measurement. The authors implement gold nanoparticles (30 nm) to benchmark iNTA against TEM, SEM, DLS and NTA. Next, the authors use polydisperse mixtures of gold nanoparticles (15, 20 and 30 nm) demonstrating that iNTA is capable to resolve these, in contrast to DLS and NTA. Finally, the authors evaluate the sensitivity and resolution of iNTA to synthetically produced liposomes and EVs obtained from cell cultures by high speed pelleting and urine. I review this manuscript mainly in relation to the applicability of the proposed method for EV research. Since NTA and interferometric detection of scattering are commonly implemented in EV research, the novelty lies in the combination of both to increase sensitivity and precision in size and RI measurements. Although I question whether this technological advance is sufficient to justify publication in Nature Methods, the approach is potentially interesting for EV research, but the work presented is preliminary, requires a more rigorous approach, further validation and understanding of the caveats in EV research for correct data interpretation.

-Although the authors benchmark iNTA for gold nanoparticles they do not implement this for EVs from cell cultures and urine samples, although these samples are much more challenging in terms of complexity induced by the presence of other extracellular particles (non-EV particles) and in terms of lower refractive index of EVs compared to gold nanoparticles.

-To demonstrate the applicability of iNTA for EVs, the authors should benchmark iNTA against TEM, NTA and DLS. The EV samples used by for iNTA have not been characterized in agreement with the current state-of-the-art (cfr. MISEV2018 guidelines), which hampers the interpretation of iNTA results described.

-The authors should consider that the samples prepared from cell culture by high speed centrifugation and the use of urine as such without any separation of EVs are not devoid of other extracellular particles (non-EV particles). For example, urine contains abundant THP complexes. Since NTA is not able to distinguish EVs from non-EV particles, such as THP complexes, it is unclear what the impact is of the results described in this manuscript related to the size, RI and protein content of EVs. A more valid approach is warranted, consisting of the preparation of high quality EV samples, that are first characterized according to the current state-of-the-art, analyzed by iNTA (using different concentrations

and size ranges (see below) to assess lower and upper detection ranges) and benchmarked against NTA, TEM and DLS. These experiments should be performed in combination with protease treatment and detergents to understand the impact on non-EV particles on iNTA measurements.

-The sensitivity, precision and detection limits of iNTA for EVs are unclear and should be further established and validated. Liposomes are filtered through a 200nm filter to exclude larger particles, indicating that iNTA is not applicable to liposomes larger in size and by extension also not applicable to larger EVs. This should be validated by preparing EVs with different size ranges for example by the use of flow field flow fractionation, which can clarify the precision of iNTA for different EV size ranges. This will also help to understand the applicability of iNTA to highly heterogeneous EV samples obtained from different biofluids.

-How is the conclusion that particles are mainly consisting of water influenced by the presence of non-EV particles that do not contain lipid membranes? What are a,b,c,d,e,f in fig 4f? Do these refer to the subgroups of EVs indicated in the manuscript text?

-p4: The authors indicate in the benchmark study using gold nanoparticles: “the accuracy of each method depends strongly on careful calibrations and consideration of various systematic effects.” How are these calibrations done for all the instruments included in the benchmark? And then more specifically, how is this calibration done for EVs?

Reviewer #3:

Remarks to the Author:

In this paper, the authors present a method called iNTA that combines microscopic tracking analysis with interferometric detection/microscopy iSCAT.

Although I have no personal experience with the method, iSCAT seems to be generally established for about ten years, with some earlier work published by the same group. In addition, NTA is commercialized by Malvern-NanoSight and used by several groups worldwide.

The claimed novelty thus depends on the combined use of both, which the authors say allows independent measurement of size and refractive index (via the scattering strength or cross-section). To this end, the authors present several calibration experiments on gold nanoparticles and polystyrene and silica colloids of 100nm diameter or less. Finally, they apply the method to a few biological samples.

iNTA and standard NTA offer the advantage of single-particle detection, which provides an inherent advantage over DLS for the characterization of polydispersity and multimodal mixtures in colloidal and nanoparticle suspensions. The downside is the more complicated sample preparation and measurement time – the authors should provide more information about their measurement protocol, reproducibility, sensitivity and the possible influence of other scattering species in the serum.

Like holographic particle sizing for larger particles (see ref 12 Lee et al.) – iSCat adds a new dimension, namely access to the scattering cross-section of the particle. The latter can be related to the particle refractive index under some assumptions. This approach breaks down however for nonspherical (larger) particles, which should be mentioned in the text. The analysis is also more complex for core-shell particles as discussed in the text. The tracking part does not suffer

from these limitations since the hydrodynamic radius is always well defined (as it is in DLS).

The iNTA method can be potentially beneficial for smaller nanoparticles and mixtures of different nanoparticle species. I am not expert enough in biology to judge whether this benefit is enormous for the case of biological specimens. The examples given by the authors in Figure 4 are not entirely convincing in my view. In the first example (Liposomes), the authors must assume the lipid refractive index for fitting the shell thickness. The second example (Extracellular vesicles from Parasites) requires several adjustable parameters, and it remains unclear how robust this fit is. For the last example (Urine) the data is all over the place and it remains unclear what additional information can be extracted from iNTA compared e.g., to DLS.

In summary, I believe this is an interesting study, scientifically sound, well written, and detailed. However, the authors should provide more context, and referees from the field of biology and nanoparticle research may be able to assess the significance that might warrant publication in Nature Methods.

Author Rebuttal to Initial comments

Reply to Referees

Kashkanova, *et al.*

Referee 2

Kashkanova et al. combine two existing methods that are routinely used in the nanoparticle field, namely interferometric detection of scattering and nanoparticle tracking analysis, to identify both size and refractive index with a higher sensitivity and precision in a single measurement. The authors implement gold nanoparticles (30 nm) to benchmark iNTA against TEM, SEM, DLS and NTA. Next, the authors use polydisperse mixtures of gold nanoparticles (15, 20 and 30 nm) demonstrating that iNTA is capable to resolve these, in contrast to DLS and NTA. Finally, the authors evaluate the sensitivity and resolution of iNTA to synthetically produced liposomes and EVs obtained from cell cultures by high speed pelleting and urine. I review this manuscript mainly in relation to the applicability of the proposed method for EV research. Since NTA and interferometric detection of scattering are commonly implemented in EV research, the novelty lies in the combination of both to increase sensitivity and precision in size and RI measurements. Although I question whether this technological advance is sufficient to justify publication in Nature Methods, the approach is potentially interesting for EV research, but the work presented is preliminary, requires a more rigorous approach, further validation and understanding of the caveats in EV research for correct data interpretation.

Our reply

We thank the referee for a careful consideration of our work and the recognition of its impact on EV research. We believe our work would be well placed in Nature Methods because we present a method for determining the size and refractive index of bioparticles in mixtures at unprecedented precision and resolution. We emphasize that iNTA is easy to use and is compatible with samples in the solution phase, i.e., without the challenges confronted in EM studies. We clearly benchmark iNTA with various well-known nanoparticle samples to demonstrate its superior performance in identifying sub-species of a mixture of nanoparticles before applying it to unknown samples. Here, we have chosen to target EV research as one of the most exciting areas of current interest and we are pleased that the referee also finds our technique interesting for advancing this area. Having focused on the performance of the method, we had indeed

not performed extensive control experiments to characterize and quantify our EV samples with independent means. We thank the referee for prompting us to do so. Our new insights have significantly strengthened our conclusions.

Referee 2

-Although the authors benchmark iNTA for gold nanoparticles they do not implement this for EVs from cell cultures and urine samples, although these samples are much more challenging in terms of complexity induced by the presence of other extracellular particles (non-EV particles) and in terms of lower refractive index of EVs compared to gold nanoparticles.

Our reply

The referee rightfully points out that the refractive index of the material plays a central role in determining the minimum detectable size. For the Leishmania exosome sample, the distribution of vesicles ranging in size from 70 to 150 nm was already observed using TEM,¹ also consistent with our measurements. As shown in Fig. 1, NTA measurements on Leishmania exosomes performed on a commercial device (ZetaView, ParticleMetriX, Germany) result in a wider distribution about a mean diameter of 150 nm with a large standard deviation of 120 nm.

We have now performed more benchmarks with state-of-the-art methods for urine samples and show that 1) TEM confirms the abundance of EVs in our urine samples (see Fig. 2a,b), 2) the size distribution of the TEM particles is consistent with the results of iNTA (see Fig. 2c), 3) Western blot measurements confirm the presence of CD81, which is expected in EVs (see inset in Fig. 2b),

4) application of detergent eliminates the particles in urine (see Fig. 4), 5) application of proteinase K eliminates the lower branch of the 2D scatter plot in the urine data (see Fig. 5), 6) Fluorescence labeling with a lipophilic dye confirms the origin of the upper branch in the urine data (see Fig. 6).

Figure 1: Size distribution of leishmania exosomes measured using a commercial NTA apparatus

Figure 2: Measurements of human urine **(a,b)** TEM measurements. Green arrows mark the EVs. The THP (UMOD) complexes are labeled with red arrows. **(c)** Comparison of size distribution histograms for TEM, NTA, and iNTA **(d)** NTA measurement **(e)** iNTA measurement performed at high laser power of 40 mW **(f)** iNTA measurement performed at low laser power of 4 mW. The dashed lines indicate the various effective RI from 1.34 in to 1.66 in increments of 0.04.

Figure 3: Characterization of liposomes extruded through 200 nm filter (a) Via DLS, showing the number-weighted distribution. **(b)** Via NTA. **(c)** Via iNTA. The green line shows a fit to Mie theory, the size histogram from iNTA is overlaid with the size histogram from NTA (gray) and DLS-record 3 (orange).

Referee 2

-To demonstrate the applicability of iNTA for EVs, the authors should benchmark iNTA against TEM, NTA and DLS. The EV samples used by for iNTA have not been characterized in agreement with the current state-of-the-art (cfr. MISEV2018 guidelines), which hampers the interpretation of iNTA results described.

Our reply

We have now followed the referee's suggestion and benchmarked iNTA against known methods. In Fig. 3, we compare the results of DLS, NTA, and iNTA for a liposome sample. The left graph shows the number-weighted distribution of three consecutive measurements of the sample. The middle graph presents a very broad intensity distribution obtained in an NTA experiment. This is to be contrasted with the high-definition plot on the right side, which results from an iNTA recording. We now include these measurements in the SI.

The Leishmania samples we used were already shown to contain exosomes exclusively.¹ We have now taken up the referee's advice and have consulted the MISEV2018 guidelines for evaluating the urine samples. We performed DLS and NTA measurements as well as TEM measurements. Furthermore, we prepared a Western Blot (WB) of the urine sample. The results are shown in Fig. 2. We note that DLS was unable to measure the sample reliably. The iNTA measurements were performed at two different powers (40 mW and 4 mW before the objective), which correspond to different sensitivities.

As the referee correctly noted, in contrast to the leishmania sample the urine sample was not ultracentrifuged, but instead prepared as described in Ref. [2]. The urine sample was not upconcentrated for TEM measurements. For WB, however, the sample was upconcentrated by a factor of about 20x. As expected, THP (also known as UMOD) was abundantly present at around 90 kDa and also CD81 was detectable in small amounts at around 25 kDa (see inset of Fig. 2b). We also stained against GAPDH but did not see any band at around 37 kDa. In a positive control measurement, using HeLa cell lysate, GAPDH was detected (data not shown).

Having seen the characteristic features of extracellular vesicles in our TEM images and considering the missing ultracentrifugation step, we confirm the presence of extracellular vesicles in our urine sample. We also performed further experiments in which an unspecific protease and various types of detergents were mixed with urine. 2% Triton, 1% Octyl-beta-Glucoside, 0.6% Sodium cholate hydrate, and 0.4% CHAPS were mixed in DPBS (1x) and added to urine 1:1. As an unspecific protease, we used proteinase K (Ambion Inc., Austin/US) at a final concentration of 200 $\mu\text{g/ml}$ and incubated it at 37°C or room temperature for 1.5 h or 2.5 h, respectively. In addition, we did negative control experiments, where urine was kept at 37°C for 1.5 h but no protease was added. Furthermore, we measured the detergent only (mixed 1:1 with DPBS). All measurements were performed over 20 minutes. The results are shown in Fig. 4. We find that almost all detectable particles were dissolved upon application of detergent.

Interestingly, only the subpopulation with an effective refractive index of 1.35 disappeared when adding proteinase K. As shown by our WB and further suggested by the polymeric structures found in the TEM images (see Fig. 2b), we identified with uromodulin at least one protein in urine that potentially could form protease- and detergent-sensitive particles.

Given that the majority of particles detected in TEM showed membranous structures and that the transmembrane protein CD81 was also identified, we believe that the protease-resistant subpopulation are extracellular vesicles.

Moreover, we note that the effective refractive index of urine EVs is close that of *Leishmania* EVs.

Finally, we performed measurements in which urine treated with proteinase was fluorescently labeled with a lipophilic dye (R18, final concentration: 2 $\mu\text{g/ml}$). The iSCAT and fluorescent measurements were performed simultaneously on two aligned and

Figure 4: Control experiments in which urine was measured in a solution with DPBS (left), in a solution with detergent mix (middle) and detergent mix alone (right). All experiments were performed over 20 minutes.

Figure 5: Control experiments in which urine was incubated at 37°C (left), was incubated with proteinase K at 37°C (middle) and incubated with proteinase K at room temperature (right). All experiments were performed over 20 minutes.

synchronized cameras. The cameras were aligned using spin-coated particles and synchronised using a signal output by the iSCAT camera as a trigger to the fluorescent camera. In order to prevent bleaching of the fluorescence, a shutter was employed to block the laser beam for 7 seconds and turn it on for 3 seconds. In addition, laser power was reduced. To collect enough fluorescence photons, the frame rate was reduced to 1 kHz with 0.92 ms exposure time. The fluorescence video was median background corrected. Afterwards, at each position of the particle a value for fluorescence was extracted by using a small Gaussian mask, similar to the iSCAT analysis. The particle fluorescence was then assigned as the median of the trajectory's five highest fluorescence values. The fluorescence of each particle was normalized by the standard deviation of the fluorescence video.

Three exemplary iSCAT and fluorescence frames are shown in Fig. 6a. Some particles are pointed out along with their fluorescence value. We note that many particles did not show fluorescence. That could be caused by a low labeling efficiency or bleaching. In Fig. 6b, we present all particles tracked in an iSCAT video, while in Fig. 6c we only show particles with a fluorescent signal larger than 2σ above the video mean. It is evident that while the number of fluorescent particles is lower, the overall shape of the distribution remains unchanged. These data are now included in the SI.

Figure 6: Measurement of fluorescently labeled urine. (a) Exemplary iSCAT and fluorescence frames. Arrows point out some of the particles together with their fluorescence values. **(b)** An iNTA scatter plot of all particles present in urine. **(c)** An iNTA scatter plot of particles that also showed a fluorescence signal at least 2σ above the mean.

Referee 2

-The authors should consider that the samples prepared from cell culture by high speed centrifugation and the use of urine as such without any separation of EVs are not devoid of other extracellular particles (non-EV particles). For example, urine contains abundant THP complexes. Since NTA is not able to distinguish EVs from non-EV particles, such as THP complexes, it is unclear what the impact is of the results described in this manuscript related to the size, RI and protein content of EVs. A more valid approach is warranted, consisting of the preparation of high quality EV samples, that are first characterized according to the current state-of-the-art, analyzed by iNTA (using different concentrations and size ranges (see below) to assess lower and upper detection ranges) and benchmarked against NTA, TEM and DLS. These experiments should be performed in combination with protease treatment and detergents to understand the impact on non-EV particles on iNTA measurements.

Our reply

We fully agree with the referee's concerns regarding the care that needs to go into sample preparation if one is to rule out unwanted nanoparticles. We would like to address this concern in three distinct points:

- 1- Following the referee's advice, we have characterized our urine samples and have presented the results in Figs. 2-6. We have verified that our preparation protocol efficiently, and to a large extent, eliminates particles other than EVs. In particular, our new iNTA investigations reveal that application of protease eliminates residual non-EV particles.
- 2- The key advantage of iNTA is that it allows one to resolve sub-species much better than other optical methods. In other words, even if there are impurities in a sample, it is very likely that iNTA will be able to identify them. A very good example concerns our new insight into the lower branch of the iNTA urine plot which attribute to a protein aggregate population. Interestingly, Fig. 7 shows that a conventional NTA measurement cannot distinguish between a raw sample and one that is cleared of this protein population upon the application of protease. Note that we

performed measurements at several gain settings of the camera for NTA to ensure that we exploited the highest sensitivity of the instrument. This figure is now included in the SI.

Figure 7: Comparison of urine measurements with iNTA and NTA. (Top row) Urine treated with proteinase K (Bottom row) Urine incubated at 37°C for 1.5 hours.

3- Our work focuses on establishing a new method that can be valuable to a wide range of studies involving bioparticles. It should be kept in mind, however, that EVs of a certain dimension and composition might have a similar iNTA signature as other bioparticles of comparable size and effective refractive index. What iNTA can do well is to characterize the distribution of size and refractive index of nanoparticles in a sample. This helps establishing high-resolution libraries and signatures for samples of different origin, e.g., for distinguishing the urine of a healthy person from that of an ill patient. Indeed, we are currently collaborating with nephrologists to establish a library for identification of various symptoms and illnesses. To present a glimpse of this exciting prospect, in Fig. 8 we compare two examples of urine samples from healthy donors with one patient with a medical condition. The sensitivity and resolution of iNTA clearly identifies differences. Considering that medical research requires very solid statistics, conclusive results of this research still need much more investigations before they can be published.

Referee 2

-The sensitivity, precision and detection limits of iNTA for EVs are unclear and should be further established and validated. Liposomes are filtered through a 200nm filter to exclude larger particles, indicating that iNTA is not applicable to liposomes larger in size and by extension also not applicable to larger EVs. This should be validated by preparing EVs with different size ranges for example by the use of flow field flow fractionation, which can clarify the precision of iNTA for different EV size ranges. This will also help to understand the applicability of iNTA to highly heterogeneous EV samples obtained from different biofluids.

Our reply

We believe, we provide a thorough discussion and analysis of the sensitivity of iNTA in Figs. 2 and 3. Furthermore, we provide a quantitative discussion of the data analysis and interpretation for liposomes of known origin in Fig. 4a,b.

These are also elaborated further in the corresponding discussions in the supplementary information.

It is important to note that there is no theoretical limit that applies to the resolution and sensitivity of iNTA, and indeed, further improvements are well within reach. In this publication, we argue and show that the current degree of added value of iNTA as compared to NTA and DLS is already substantial enough for sharing our results with the community of scientists who would benefit from them. We hope that the revised manuscript and SI as well as the discussion we provide in this reply letter has clarified these points further.

Liposomes were filtered through a 200 nm filter in our experiment because we wanted to make sure that we examined unilamellar vesicles as a well-controlled nontrivial sample.³ iNTA can also easily be applied to particles of larger sizes. In fact, as in most

Figure 8: iNTA analysis of urine sample from two healthy donors and a donor with antibody-mediated kidney transplant rejection.

other measurements, a larger signal becomes easier to analyze. However, the superior performance and novelty of iNTA lie in its applicability to smaller particles since other techniques such as NTA or holographic NTA can also perform well

for larger particles. Here, it is to be noted that the generalized Mie theory, which has been used to analyze our EV data, is capable of handling larger particles. The only technical issue that has to be kept in mind is that when very polydisperse samples are measured, the measurements may need to be performed at different laser powers to avoid saturation of the camera by the large particles. We address this issue in Fig. S2 and S4 in the supplementary information.

To provide direct evidence that iNTA is capable of working with larger particles, we measured commercially available beads with nominal diameters of 165 nm and 500 nm. Figure 9 displays the outcome. We remark that the manufacturer specifies a large spread in the particle sizes, which we confirmed with SEM investigations. The extracted refractive index is also well within the literature values for silica for both samples.

Referee 2

-How is the conclusion that particles are mainly consisting of water influenced by the presence of non-EV particles that do not contain lipid membranes?

Our reply

The conclusion that the particles mainly consist of water stems from the fact that all organic material has a larger refractive index than water ($RI=1.334$). This conclusion holds regardless of whether the particle is an EV or not. As we show in the SI, the effective refractive index for particles from the lower subgroup in the urine plot is below 1.35. Assuming these particles are protein aggregates ($RI = 1.58$) would imply less than 7% of the aggregate by volume is actually protein, and that the rest is water. This would roughly be consistent with the findings of Ref. [4].

Referee 2

-What are a,b,c,d,e,f in fig 4f? Do these refer to the subgroups of EVs indicated in the manuscript text?

Our reply

We thank the referee for pointing out this mistake. The letters were left in the figure from the previous iteration of the draft and have now been removed.

Referee 2

-p4: The authors indicate in the benchmark study using gold nanoparticles: the accuracy of each method depends strongly on careful calibrations and consideration of various systematic effects. How are these calibrations done for all the instruments included in the benchmark? And then more specifically, how is this calibration done for EVs?

Figure 9: Plots of contrast and refractive index vs. size for 165 nm silica beads and 500 nm silica beads.

Our reply

A careful calibration of an iNTA setup involves measuring the temperature of the sample and the pixel size of the camera (as is necessary for NTA). We measured the temperature using an infrared camera. The camera pixel size was calculated from our knowledge of the optical components and measured using a specifically fabricated calibration sample with lines spaced by a given distance. To render the calibration self-consistent, one uses particles of known size and refractive index, as described in the manuscript. Further studies on unknown samples can then be analyzed based on these measured parameters.

The calibrations for DLS and NTA were performed by the manufacturers. What we meant to convey with that sentence is that the mean values of the distributions are more prone to error than their widths. We have now rephrased this sentence to ensure that it has a clear message.

We thank referee 2 for his/her valuable and constructive criticism of our results, which prompted us to present further evidence for our EV work. This has clearly improved the last section of the paper.

Referee 3

In this paper, the authors present a method called iNTA that combines microscopic tracking analysis with interferometric detection/microscopy iSCAT. Although I have no personal experience with the method, iSCAT seems to be generally established for about ten years, with some earlier work published by the same group. In addition, NTA is commercialized by Malvern-NanoSight and used by several groups worldwide. The claimed novelty thus depends on the combined use of both, which the authors say allows independent measurement of size and refractive index (via the scattering strength or cross-section). To this end, the authors present several calibration experiments on gold nanoparticles and polystyrene and silica colloids of 100nm diameter or less. Finally, they apply the method to a few biological samples.

Our reply

We thank the referee for a careful consideration of our work. The novelty of the work is, indeed, in the combination of iSCAT and NTA and the demonstration that this provides an unprecedented set of advantages (smaller particles, more resolution in deciphering particle mixtures, more precision in determining refractive indices) that is highly desirable in several current research areas.

Referee 3

iNTA and standard NTA offer the advantage of single-particle detection, which provides an inherent advantage over DLS for the characterization of polydispersity and multimodal mixtures in colloidal and nanoparticle suspensions. The downside is the more complicated sample preparation and measurement time the authors should provide more information about their measurement protocol, reproducibility, sensitivity and the possible influence of other scattering species in the serum.

Our reply

It is true that single-particle and single-molecule experiments are usually more challenging than ensemble experiments. However, the measurement procedure of NTA and iNTA are both straightforward and user-friendly. In particular, the sample preparation is quite simple. A drop of liquid containing particles of interest is placed on a cover glass. Any kind of cell culture imaging dish (e.g. ibidi dish, or μ -slide) can also be used instead. A cover is used to prevent evaporation. In fact, this procedure is essentially the same for all three methods, DLS, NTA and iNTA with the mere difference that the samples for the latter two methods need to be diluted.

The measurement time in all methods is directly determined by the size of the signal and by the desired precision and accuracy. Thus, one cannot easily attribute "a measurement time": the weaker the signal or the higher the desired resolution, the longer is the data collection time, as in for any other method. For more concentrated monodisperse samples, a few seconds of measurement is sufficient for iNTA. For example, the data presented in Fig. 2d of the main text was recorded over 5 seconds. For more complex polydisperse sample and for samples with low particle concentration, longer measurements are helpful. The data presented in Fig. 3 of the main text was recorded over 10-20 minutes. To demonstrate the evolution of the data quality as a function of the measurement time, we show the outcome of the first 1 or 5 minutes of the measurement presented in Fig. 3c of the manuscript in Fig. 10. A comparison of Fig. 10b and Fig. 3c of the manuscript clearly shows that the improvement obtained between 5 min and 20 min is, in fact, incremental for deciphering the three populations. However, the longer the measurements, the more reliably can one recognize weak features, which might be, e.g., indicative of a disease or rare impurity.

We note that NTA has been engineered as a user-friendly product by several manufacturers. iNTA can also be automated for high throughput use and analysis in a similar fashion. We are currently working on these technical steps.

Figure 10: Measurement of a mixture of 15 nm, 20 nm, and 30 nm GNPs for different measurement times.

Figure 11: Size and third root of iSCAT contrast for twenty 30 nm GNP samples measured over the course of two weeks.

The sensitivities of NTA and iNTA are limited by the visibility of particles. Here, the superior signal of iSCAT over dark-field microscopy provides a decisive advantage for iNTA. Our currently achieved sensitivity can be read in several of the plots, e.g., 50 nm EVs, or 10 nm gold nanoparticles. However, considering that iSCAT has been demonstrated to detect individual unlabelled proteins in stationary samples, it is foreseeable that one could push the sensitivity limit of iNTA further than what we have demonstrated in our current work.

To address the question of reproducibility, we show the results of twenty measurements of 30 nm GNP samples conducted over the course of two weeks in Fig. 11. The mean value of size is 33.8 nm with standard deviation of 0.4 nm. The mean value of the third root of the contrast is 0.69 with standard deviation of 0.01. We now include these plots in the SI.

Referee 3

Like holographic particle sizing for larger particles (see ref 12 Lee et al.) iSCat adds a new dimension, namely access to the scattering cross-section of the particle. The latter can be related to the particle refractive index under some assumptions. This approach breaks down however for nonspherical (larger) particles, which should be mentioned in the text. The analysis is also more complex for core-shell particles as discussed in the text. The tracking part does not suffer from these limitations since the hydrodynamic radius is always well defined (as it is in DLS).

Our reply

We agree with referee's assessment. We emphasize that it is the high signal-to-noise ratio (SNR) of iSCAT that makes the two-dimensional scatter plots in iNTA more informative than the pioneering work in Ref. 12. This high SNR is exactly what makes both our diffusion measurements and our contrast measurements more precise, providing a better resolution in determining sub-populations or complex distributions.

The referee is also right in pointing out the difficulty of *all* optical methods, i.e. DLS, NTA and iNTA, in that they assume spherical particles. It is indeed known from generalized Mie and scattering theory that the scattering cross section also depends on the shape/form factor, thus, making the signal polarization dependent. By choosing a circular polarization in illumination, we minimize this dependence and obtain an *effective* refractive index. This approach is clearly an approximation that should be kept in mind when analyzing the data. For example, it has been pointed out that for protein aggregates of arbitrary shapes the effective refractive index is smaller than the protein refractive index.⁴ iNTA does not aim to fully characterize each individual nanoparticle. The valuable usage of the method, as in NTA and DLS, is to 1) obtain quantitative data if the shape and make of the particles are known, 2) characterize mixtures of unknown samples in a way to establish a high-resolution signature that can be used to distinguish anomalies, e.g. when a disease sets in (see Fig. 8).

Regarding iNTA's ability to detect larger particles: there is nothing that prevents iNTA from obtaining the same data quality as in holographic NTA since the signal in both methods relies on interference between a reference and the scattered light. In our work, we have focused on smaller particles because this is where holographic NTA has not delivered quantitative information. In Fig. 9 we present iNTA measurements on silica beads with nominal diameters of 170 nm and 500 nm. We now make a reference to this issue in last section (Discussion and Outlook) of the manuscript.

Referee 3

The iNTA method can be potentially beneficial for smaller nanoparticles and mixtures of different nanoparticle species. I am not expert enough in biology to judge whether this benefit is enormous for the case of biological specimens. The examples given by the authors in Figure 4 are not entirely convincing in my view. In the first example (liposomes), the authors must assume the lipid refractive index for fitting the shell thickness. The second example (extracellular vesicles from parasites) requires several adjustable parameters, and it remains unclear how robust this fit is. For the last example (urine) the data is all over the place and it remains unclear what additional information can be extracted from iNTA compared e.g., to DLS.

Our reply

Judging the novelty of interdisciplinary research is indeed often a challenge. Our strategy for presenting a rigorous discussion of the method together with a compelling potential for biological applications has been as follows: We start

in Fig. 2 by presenting a direct benchmark of iNTA against four other established methods. To have a quantitative comparison, we take a *well-known monodisperse* sample of gold nanoparticles. To show the superior performance of iNTA, we choose the smallest particle that NTA could also analyze. Next, in Fig. 3, we demonstrate the performance of iNTA on *well-known polydisperse* samples. Here, we start with a comparison with DLS and iNTA in parts (a-c) and then present several other samples that would not be resolved by the other two methods. In Fig. 4, we show the application to layered particles. Again, we start with a sample that is well-known, namely single-walled liposomes, so that we can establish the credibility of our analysis. Considering that we produce these liposomes ourselves from lipids of choice, we can assume a value for their refractive index. In Fig. 4b of the manuscript, we show that the refractive index of the shell (n_{sh}) varies with the assumed thickness of the shell (t_{sh}) and, most importantly, that the refractive index of the contents (n_{in}) is independent of the choice of the two and equal to the refractive index of water, as expected. We have rephrased this passage in the revised manuscript to make it clearer that our work on liposomes serve to set the quantitative ground for the sensitive analysis of shelled nanoparticles. We also clarify the confusion about the need to assume the refractive index of the lipids.

In a next step, we examine Leishmania exosomes, which have been previously detected and analyzed using electron microscopy. Here, however, we do not know the lipid types and the details of the particles. Our analysis shows that they have higher inner refractive index, which is also independent of the choices of n_{sh} and t_{sh} . Figure S7 shows that neither liposome nor leishmania exosome data can be fitted with a solid sphere model.

As a last step, we wanted to apply our method to a challenging unknown sample with potentially high impact in medical diagnostics and fundamental cell biology research. In particular, we want to emphasize that our method provides characteristic signatures with much more detail than other methods. In that sense, the “all-over-ness” of the data was a desired feature because NTA reports in the literature could not resolve this level of detail.

We have now done extensive characterizations of the urine sample in response to the comments of referee 2 to clarify which particles are extracellular vesicles. In this process, we have shown that the lower branch of the data in the urine scatter plot is sensitive to protease activity and most likely represents protein aggregates. This discovery nicely presents the advantage of iNTA: In the original submission, we had prepared our samples according to the protocol used by van der Pol *et al.* in Ref. [2], which presents an NTA study of urine. However, because NTA does not resolve the two branches of the data, it could not hint to the existence of two different particle species. In our new Fig. 4e, we now emphasize this advantage of iNTA. The particles in the upper branch were fitted with a shell model, as shown in the main manuscript.

In summary, the richness of the data is a signature of the sample. We are currently collaborating with nephrologists to establish a library for identification of various symptoms and illnesses (see Fig. 8). It should be kept in mind, however, that medical research requires very solid statistics so that the result of this kind of research might need a long time to be published.

Referee 3

In summary, I believe this is an interesting study, scientifically sound, well written, and detailed.

However, the authors should provide more context, and referees from the field of biology and nanoparticle research may be able to assess the significance that might warrant publication in Nature Methods.

Our reply

We thank the referee for the overall positive assessment of our work. We hope that our revisions and responses address the concerns raised by the referees.

References

- 1 Leitherer, S. *et al.* Characterization of the protein tyrosine phosphatase LmPRL-1 secreted by leishmania major via the exosome pathway. *Infection and Immunity* **85** (2017).
- 2 van der Pol, E., Coumans, F. A. W., Sturk, A., Nieuwland, R. & van Leeuwen, T. G. Refractive index determination of nanoparticles in suspension using nanoparticle tracking analysis. *Nano Letters* **14**, 6195–6201 (2014).
- 3 Scott, H. L. *et al.* On the mechanism of bilayer separation by extrusion, or why your LUVs are not really unilamellar. *Biophysical Journal* **117**, 1381–1386 (2019).
- 4 Odete, M. A. *et al.* The role of the medium in the effective-sphere interpretation of holographic particle characterization data. *Soft Matter* **16**, 891–898 (2020).

Decision Letter, first revision:

Dear Vahid,

Thank you for submitting your revised manuscript "Precision size and refractive index analysis of weakly scattering nanoparticles in polydispersions." (NMETH-A47001A-Z). It has now been seen by the original referees and their comments are below. The reviewers find that the paper has improved in revision, and therefore we'll be happy in principle to publish it in Nature Methods, pending minor revisions to satisfy the referees' final requests and to comply with our editorial and formatting guidelines.

We ask that you address the remaining concerns of referee 2, which include modifications to the text as well as a proteinase K control experiment.

TRANSPARENT PEER REVIEW

Nature Methods offers a transparent peer review option for new original research manuscripts submitted from 17th February 2021. We encourage increased transparency in peer review by publishing the reviewer comments, author rebuttal letters and editorial decision letters if the authors agree. Such peer review material is made available as a supplementary peer review file. **Please state in the cover letter 'I wish to participate in transparent peer review' if you want to opt in, or 'I do not wish to participate in transparent peer review' if you don't.** Failure to state your preference will result in delays

in accepting your manuscript for publication.

Thank you again for your interest in Nature Methods Please do not hesitate to contact me if you have any questions.

Sincerely,
Rita

Rita Strack, Ph.D.
Senior Editor
Nature Methods

ORCID

Reviewer #2 (Remarks to the Author):

In the revised version of the manuscript the authors present new data. These new data add to the potential value of iNTA for extracellular vesicles demonstrating that in contrast to other methods, it allows visualization of extracellular vesicles from other particles using appropriate control experiments (e.g. detergent and protease treatments). Nevertheless some comments remain:

-Clarify the number of samples that have been used to establish the iNTA plots. Or in the figure legends or in the main text. It is clearly indicated how many trajectories are evaluated, but please clarify how many samples have been analyzed. For example the iTNA plots for Leishmania and urinary extracellular vesicles. How many biological and technical replicates did the authors perform? Although the authors demonstrate reproducibility on gold nanoparticles, this information is still lacking for EVs (urine and parasite)?

-Ideally, proteinase K treatment requires a control to demonstrate that extracellular vesicles are still intact upon treatment (e.g. western blot analysis for a cytosolic protein and a membrane protein).

-Manuscript should consider adequately the limitations of the study as well in the discussion/conclusion section. For example, how applicable is iNTA to more complex samples than e.g. blood? Etc. What is the immediate applicability/implementation of iNTA? How does the measurement time differ between the different evaluated platforms and iNTA? Does it allow high-throughput analysis of samples (compared to other evaluated platforms)?

-Western blot on CD81 and UMOD is not properly presented. Please show whole blot membranes with size markers.

-The authors indicate the parasite samples were already shown to contain “exosomes” exclusively. One of the challenges in extracellular vesicle research is to exclusively obtain extracellular vesicles from biofluids. Better would be to indicate that those samples are enriched in extracellular vesicles.

-Both terms exosomes and extracellular vesicles are used. Please replace exosomes with extracellular vesicles throughout manuscript, figures, figure legends and supplementary information.

-Submit experimental parameters to the EV-TRACK knowledgebase.

-The main text update on parasite and urine EVs experiments is not very clear and does not read fluently and lacks the appropriate information required for the reader to understand the experiment and the added value of the results.

Reviewer #3 (Remarks to the Author):

The reply by the authors to the referee's comments is satisfactory in my view. In addition, the authors made a few revisions that, in my opinion, improved the manuscript. As a result, the paper is technically sound, well written, and presented. Thus Nature Methods could publish it, and we'll find out about the impact in practice over the following years.

Author Rebuttal, first revision:

Reply to Referees

Kashkanova, *et al.*

Referee 2

In the revised version of the manuscript the authors present new data. These new data add to the potential value of iNTA for extracellular vesicles demonstrating that in contrast to other methods, it allows visualization of extracellular vesicles from other particles using appropriate control experiments (e.g. detergent and protease treatments). Nevertheless some comments remain:

Our reply

We thank the referee for the positive assessment of the new experiments.

Referee 2

-Clarify the number of samples that have been used to establish the iNTA plots. Or in the figure legends or in the main text. It is clearly indicated how many trajectories are evaluated, but please clarify how many samples have been analyzed. For example the iNTA plots for Leishmania and urinary extracellular vesicles. How many biological and technical replicates did the authors perform? Although the authors demonstrate reproducibility on gold nanoparticles, this information is still lacking for EVs (urine and parasite)?

Our reply

Each iNTA plot presents data acquired from a single sample via a single measurement. This information has now been added in the figure caption. We did check that the repeated measurements of the same sample, as well as measurements of the similarly prepared samples result in very similar and equivalent results. Several of those measurements are now shown in the SI and in the Fig. 1 below. We note that we have established the reproducibility, sensitivity and accuracy of our optical measurement by using well-defined samples. As a result, we can attribute any observed deviations among unknown samples to their inhomogeneities.

Referee 2

-Ideally, proteinase K treatment requires a control to demonstrate that extracellular vesicles are still intact upon treatment (e.g. western blot analysis for a cytosolic protein and a membrane protein).

Our reply

We were quite confident that EVs were still intact after the treatment because the particles that were left (giving rise to the upper branch of the iNTA plot in Fig. 4e) disappeared upon application of detergent. Nevertheless, we have followed the referee's advice and performed a western blot on the urine samples with and without added proteinase K. The results presented in the SI confirm that a correct application of proteinase K avoids damage to the EVs.

Referee 2

-Manuscript should consider adequately the limitations of the study as well in the discussion/conclusion section. For example, how applicable is iNTA to more complex samples than e.g. blood? Etc. What is the immediate applicability/implementation of iNTA? How does the measurement time differ between the different evaluated platforms and iNTA? Does it allow high-throughput analysis of samples (compared to other evaluated platforms)?

Figure 1: Reproducibility check for biological samples. Each column shows one of the measured sample types. Rows 1 and 2 show measurements of the same sample performed on different days, while rows 2 and 3 show the measurement of different samples prepared on the same day. Solid line indicates the fit to the data, while dashed line indicates the fit to the data shown in the main text.

Our reply

We now address some aspects of the technical performance/limitations of iNTA in the discussion section and in the SI.

We believe it is not possible to make a simple statement about the kind of sample that can be or cannot be studied by iNTA. We state explicitly that "The high-resolution 2D scatter plots presented in this work serve to detect slight variations and anomalies among different samples." The wealth of information that one might obtain varies strongly on the make of the sample. Our detailed studies presented in Figs. 2,3,4 give the reader a very good insight into the resolution and sensitivity that he/she might expect in a specific application.

Referee 2

-Western blot on CD81 and UMOD is not properly presented. Please show whole blot membranes with size markers.

Our reply

We thank the referee for pointing this out. This has now been edited.

Referee 2

-The authors indicate the parasite samples were already shown to contain "exosomes" exclusively. One of the challenges in extracellular vesicle research is to exclusively obtain extracellular vesicles from biofluids. Better would be to indicate that those samples are enriched in extracellular vesicles.

Our reply

We thank the referee for pointing this out. This has now been edited.

Referee 2

-Both terms exosomes and extracellular vesicles are used. Please replace exosomes with extracellular vesicles throughout manuscript, figures, figure legends and supplementary information.

Our reply

We thank the referee for pointing this out. This has now been edited.

Referee 2

-Submit experimental parameters to the EV-TRACK knowledgebase.

Our reply

You may access and check the submission of experimental parameters to the EV-TRACK knowledgebase via the following URL: <http://evtrack.org/review.php>. Please use the EV-TRACK ID (EV220073) and the last name of the first author (Kashkanova) to access our submission. This information is now included in Data Availability Statement.

Referee 2

-The main text update on parasite and urine EVs experiments is not very clear and does not read fluently and lacks the appropriate information required for the reader to understand the experiment and the added value of the results.

Our reply

We have now rewritten the sections to be more clear.

Referee 3

The reply by the authors to the referee's comments is satisfactory in my view. In addition, the authors made a few revisions that, in my opinion, improved the manuscript. As a result, the paper is technically sound, well written, and presented. Thus Nature Methods could publish it, and we'll find out about the impact in practice over the following years.

Our reply

We thank the referee for the positive assessment of our work. We are currently involved in several collaborations and hope to demonstrate the impact of iNTA to the scientific community very soon.

Final Decision Letter:

Dear Vahid,

I am pleased to inform you that your Article, "Precision size and refractive index analysis of weakly scattering nanoparticles in polydispersions.", has now been accepted for publication in Nature Methods. Your paper is tentatively scheduled for publication in our May print issue, and will be published online prior to that. The received and accepted dates will be Sep 1, 2021 and March 18, 2022. This note is intended to let you know what to expect from us over the next month or so, and to let you know where to address any further questions.

In approximately 10 business days you will receive an email with a link to choose the appropriate

publishing options for your paper and our Author Services team will be in touch regarding any additional information that may be required.

Your paper will now be copyedited to ensure that it conforms to Nature Methods style. Once proofs are generated, they will be sent to you electronically and you will be asked to send a corrected version within 24 hours. It is extremely important that you let us know now whether you will be difficult to contact over the next month. If this is the case, we ask that you send us the contact information (email, phone and fax) of someone who will be able to check the proofs and deal with any last-minute problems.

If, when you receive your proof, you cannot meet the deadline, please inform us at rjsproduction@springernature.com immediately.

Once your manuscript is typeset and you have completed the appropriate grant of rights, you will receive a link to your electronic proof via email with a request to make any corrections within 48 hours. If, when you receive your proof, you cannot meet this deadline, please inform us at rjsproduction@springernature.com immediately.

Once your paper has been scheduled for online publication, the Nature press office will be in touch to confirm the details.

Content is published online weekly on Mondays and Thursdays, and the embargo is set at 16:00 London time (GMT)/11:00 am US Eastern time (EST) on the day of publication. If you need to know the exact publication date or when the news embargo will be lifted, please contact our press office after you have submitted your proof corrections. Now is the time to inform your Public Relations or Press Office about your paper, as they might be interested in promoting its publication. This will allow them time to prepare an accurate and satisfactory press release. Include your manuscript tracking number NMETH-A47001B and the name of the journal, which they will need when they contact our office.

About one week before your paper is published online, we shall be distributing a press release to news organizations worldwide, which may include details of your work. We are happy for your institution or funding agency to prepare its own press release, but it must mention the embargo date and Nature Methods. Our Press Office will contact you closer to the time of publication, but if you or your Press Office have any inquiries in the meantime, please contact press@nature.com.

If you are active on Twitter, please e-mail me your and your coauthors' Twitter handles so that we may tag you when the paper is published.

Please note that *Nature Methods* is a Transformative Journal (TJ). Authors may publish their research with us through the traditional subscription access route or make their paper immediately open access through payment of an article-processing charge (APC). Authors will not be required to make a final decision about access to their article until it has been accepted. Find out more about Transformative Journals

To assist our authors in disseminating their research to the broader community, our SharedIt initiative provides you with a unique shareable link that will allow anyone (with or without a subscription) to read the published article. Recipients of the link with a subscription will also be able to download and print the PDF. As soon as your article is published, you will receive an automated email with your shareable link.

Please note that you and your coauthors may order reprints and single copies of the issue containing your article through Nature Research Group's reprint website, which is located at <http://www.nature.com/reprints/author-reprints.html>. If there are any questions about reprints please send an email to author-reprints@nature.com and someone will assist you.

Best regards,
Rita